# Transport parameterization of the Polar SWIFT model (version 2)

Ingo Wohltmann[1], Daniel Kreyling[1], and Ralph Lehmann[1]

[1]Alfred Wegener Institute for Polar and Marine Research, Potsdam, Germany

**Correspondence:** I. Wohltmann (ingo.wohltmann@awi.de)

**Abstract.** The Polar SWIFT model is a fast scheme for calculating the chemistry of stratospheric ozone depletion in the polar vortex in winter. It is intended for use in General Circulation Models (GCMs) and Earth System Models (ESMs) to enable the simulation of interactions between the ozone layer and climate, when a full stratospheric chemistry scheme is computationally too expensive. In addition to the simulation of chemistry, ozone has to be transported in the GCM. As an alternative to the general schemes for the transport and mixing of tracers in the GCMs, a parameterization of the transport of ozone can be used in order to obtain the total change of ozone as the sum of the change by transport and by chemistry. A benefit of this approach includes the easy and self-contained coupling to a GCM. Another advantage can be that a transport parameterization based on reanalysis data and measurements can avoid deficiencies in the representation of transport in the GCMs, like deficits in the representation of the Brewer-Dobson circulation caused by the gravity wave parameterization. Hence, we present a transport parameterization for the Polar SWIFT model that simulates the change of vortex-averaged ozone by transport in a fast and simple way without the need for a complex transport scheme in the GCM.

## 1 Introduction

The importance of interactions between climate change and the ozone layer has long been recognized (e.g., Thompson and Solomon, 2002; Rex et al., 2006; Nowack et al., 2015). Hence, it is desirable to account for these interactions in climate models. Since a full stratospheric chemistry scheme is computationally very expensive, several fast schemes for ozone chemistry, like the Cariolle scheme (Cariolle and Déqué, 1986; Cariolle and Teyssèdre, 2007), CHEM2D-OPP (McCormack et al., 2006), BMS scheme (Monge-Sanz et al., 2011, 2022) or Linoz (McLinden et al., 2000; Hsu and Prather, 2009), have been developed. Many of these schemes were originally designed only for extrapolar ozone (e.g., Cariolle and Déqué, 1986; McCormack et al., 2006) or only include a simplified approach to model polar ozone depletion (e.g., Cariolle and Teyssèdre, 2007; Hsu and Prather, 2009). Polar SWIFT is intended to add a more sophisticated polar ozone chemistry scheme to GCMs (Wohltmann et al., 2017). So far, Polar SWIFT and the transport parameterization have been implemented into the ECHAM6 climate model (Romanowsky et al., 2019), the AFES4.1 climate model (publication for the AFES version using SWIFT under preparation, for a general description of AFES, see Ohfuchi et al., 2004; Enomoto et al., 2008; Kuwano-Yoshida et al., 2010), and the ICON-NWP model 2.6.3 (publication under preparation, for a general description, see Zängl et al., 2015).

As an alternative to the general schemes for the transport and mixing of tracers in the GCMs, a parameterization of the transport of ozone can be used. A benefit of this approach includes the easy and self-contained coupling to a GCM. Another

advantage can be that a transport parameterization based on reanalysis data and measurements can avoid deficiencies in the representation of transport in the GCMs, like deficits in the representation of the Brewer-Dobson circulation caused by the gravity wave parameterization or excessive mixing. A transport parameterization based on reanalysis data and measurements

may actually perform more realistically and lead to better results than the transport of the GCM in these cases. For example, we implemented tracer transport for Polar SWIFT in ECHAM6 and found that the tracer transport of ECHAM6 overestimated the ozone concentrations inside the vortex, especially in the southern polar vortex. Actually, the results obtained by the transport parameterization were an improvement over the version with tracer transport. A reason for the performance of the tracer transport may be the overestimation of horizontal transport, which is a known issue in ECHAM6 (Stevens et al., 2013).

Polar SWIFT simulates the chemical evolution of the vortex-averaged mixing ratios of key species that are involved in polar ozone depletion by solving a set of coupled differential equations for these species on a small number of vertical levels (Wohltmann et al., 2017). That is, only a single value is computed per level. The model includes the four prognostic variables $ClONO_2$, $HCl$, $HNO_3$, and $O_3$. Only $O_3$ is returned to the GCM as input for the radiation module.

The transport parameterization computes only one vortex-averaged value for the change of ozone by transport per level.

Then, the simulated total vortex-averaged change in ozone is calculated as the sum of the vortex-averaged change from the transport parameterization and the vortex-averaged change by chemistry from Polar SWIFT. The transport parameterization includes a "constant change" term that just adds a constant amount of ozone per time step of the model, and a temperature-dependent term that considers the variability in ozone transport caused by variability in the Brewer-Dobson circulation. The only input variable from the GCM that is needed for the transport parameterization is the vortex-averaged temperature.

$ClONO_2$, $HCl$ and $HNO_3$ are not included in the transport parameterization. Tests with a transport parameterization for these species showed that the changes induced for $O_3$ were small.

Only a few variables have to be passed from the GCM to Polar SWIFT via an interface routine. These are temperature, potential vorticity and the model grid. Potential vorticity is used to determine the location of the polar vortex. Vortex-averaged temperature is then calculated from the temperature field and vortex location. For the initialization of Polar SWIFT at the

start of each winter, the $O_3$ climatology of the GCM is passed. $ClONO_2$, $HCl$ and $HNO_3$ are initialized from climatologies that are part of SWIFT. Polar SWIFT calculates the chemical tendency of vortex-averaged ozone at each level based on these variables with a typical time step of 24 h and adds the tendency to the current vortex-averaged ozone value. Then, the transport parameterization adds the change by transport to the vortex-averaged ozone value. The final vortex-averaged ozone value at each level is assigned to every GCM grid point at this level inside the vortex. The updated $O_3$ field is then passed back to the

radiation module of the GCM.

Outside of the polar vortex, the values of the internal $O_3$ climatology of the GCM, which can vary with season, are used as input for the radiation module. Tracers are not advected outside the polar vortex. There is no interpolation applied between the two domains, since the edge of the polar vortex often forms a strong barrier between air masses and strong gradients in species concentrations are common.

We recommend to only apply the transport parameterization in a GCM when a stable vortex exists. In this paper, we use a minimum size of 15 million $km^2$ at 54 hPa as a criterion for a stable vortex.

The transport parameterization is based on a fit to the transport of ozone modelled in the global Lagrangian ATLAS Chemistry and Transport Model (see Wohltmann and Rex, 2009, for a detailed description) for many different winters in the northern and southern hemisphere (similar runs are used in Wohltmann et al., 2017). These ATLAS-SWIFT runs use the full transport and mixing scheme of the ATLAS model, while the detailed stratospheric chemistry scheme of the ATLAS model is replaced by the simplified Polar SWIFT model to obtain self-consistent results.

Section 2 gives some definitions and describes the ATLAS-SWIFT model runs. Section 3 presents the equations of the transport parameterization and the rationale behind the parameterization. Section 4 shows validation results compared to the full transport model of ATLAS and Microwave Limb Sounder (MLS) measurements. Section 5 contains the conclusions.

## 2 Prerequisites

### 2.1 Definitions

SWIFT and the transport parameterization are based on 5 levels at $69.66111\,\mathrm{hPa}$, $54.03643\,\mathrm{hPa}$, $41.59872\,\mathrm{hPa}$, $31.77399\,\mathrm{hPa}$ and $24.07468\,\mathrm{hPa}$ (rounded values are used in the text in the following), which roughly encompass the vertical range in which ozone depletion is observed (see Wohltmann et al., 2017, for the choice of levels). In order to derive the parameterization, these levels are extended into adjacent layers centered at these levels. The vertical distance of the levels is about 2 km and is consistent with the typical scale on which profiles of vortex-averaged temperature, ozone, ozone depletion and descent vary (see, e.g., Wohltmann et al., 2020). The vortex edge is assumed at $\pm 36$ PVU potential vorticity at 475 K for the northern and southern hemisphere, respectively. The vortex edge criterion is extended to other altitudes than 475 K by the modified potential vorticity of Lait (1994).

### 2.2 Model runs of the ATLAS-SWIFT model

The ATLAS-SWIFT model runs are driven by meteorological data from the European Centre of Medium-Range Weather Forecasts (ECWMF) ERA5 reanalysis (provided on a $1.125^{\circ}$ x $1.125^{\circ}$ horizontal grid, 3 h temporal resolution, 137 model levels) (Hersbach et al., 2020). The model uses a hybrid vertical coordinate that is identical to a pure potential temperature coordinate for a pressure smaller than 100 hPa. Diabatic heating rates from ERA5 are used to calculate vertical motion. The vertical range of the model domain is 350–1900 K and the horizontal resolution of the model is 150 km (mean distance between air parcels).

The runs are set up in a way similar to the setup of the runs used in Wohltmann et al. (2017): At the start of the model run, $O_3$, $Cl_y$, HCl, $HNO_3$ and $ClONO_2$ are initialized with seasonal climatologies that are interpolated to the start date. For $O_3$ and $HNO_3$, a seasonal climatology based on all available data of the MLS satellite instrument (e.g., Livesey et al., 2020) from the years 2005–2011 is used (i.e., which is a function of the month of year, with data from all years averaged). $Cl_y$, $ClONO_2$ and HCl are taken from a seasonal climatology derived from ATLAS runs of the years 2005 and 2006 with the full chemistry model. As many species as possible have been initialized with MLS measurements, however $ClONO_2$ observations

do not provide enough coverage. That is why we have decided to use only ATLAS values for chlorine species, since that helps guarantee consistency of HCl and $ClONO_2$ with $Cl_y$. The full transport scheme of ATLAS is used to transport the species.

Polar SWIFT is implemented in ATLAS by adding the vortex-averaged chemical rate of change of ozone calculated by Polar SWIFT for a given layer to the ozone value of every air parcel inside the vortex and inside this layer. Note that this means that the ozone field does still vary across the vortex because of the initialization and ozone changes induced by transport. The same is done for the other simulated species HCl, $ClONO_2$ and $HNO_3$. The vortex-averaged mixing ratios of these species, which are needed as input at the start of every time step, are obtained by averaging over all air parcels inside the vortex in each

layer. Outside of the polar vortex, $O_3$, $Cl_y$, HCl, $HNO_3$ and $ClONO_2$ are reinitialized every day by interpolating the seasonal climatologies also used for initialization to the current time step.

Simulations of the Arctic winters 1979/1980–2020/2021 and the Antarctic winters 1980–2021 are performed. For every winter and hemisphere, a new run is started that is initialized with species mixing ratios from the same MLS and ATLAS climatologies that are described above (i.e., the same initial species concentrations in every year). Runs start on 1 November

and end on 31 March in the northern hemisphere and start on 1 May and end on 30 November in the southern hemisphere. The long-term change in the chlorine loading of the stratosphere is considered by multiplying the $Cl_y$, HCl and $ClONO_2$ values by a number obtained by dividing the Equivalent Effective Stratospheric Chlorine (EESC, Newman et al., 2007) of the given year by the EESC of the year 2000.

## 3  The transport parameterization

We will first give the equation for the parameterization and then discuss the calculation of the fit parameters, the rationale behind the choice of the parameterization and the derivation of the equation in Sections 3.1 to 3.3. The parameterization is based on vortex averages in the vertical layers of SWIFT to comply with the formulation of the chemistry in the Polar SWIFT model. The final equation for the change of the vortex-averaged ozone mixing ratio by transport $\Delta O_{3,GCM,transport}$ in the GCM in a time step $\Delta t$ and in a given layer and hemisphere is

$$\Delta O_{3,GCM,transport} = c_T(\Delta T_{GCM} - \Delta T_R) + c_{const}\Delta t \qquad (1)$$

We base the temperature-dependent term on an input variable that is readily available from the GCM, and that is the change of vortex-averaged temperature $\Delta T_{GCM}$ in the GCM in the layer within the time step $\Delta t$. $\Delta T_R$ is the same for the change in radiative equilibrium temperature. $\Delta T_R$ is based on a fixed lookup table of values in each layer and only depends on the day of year (see Section 3.3 for details, the lookup table for $\Delta T_R$ can be found in the source code, see link to Zenodo repository in

the data availability section).

$c_T$ and $c_{const}$ are fit coefficients for the temperature-dependent term and the "constant change" term derived from the ATLAS-SWIFT runs (see Section 3.1). The fit coefficients can be found in Table 1. The actual implementation in the ECHAM, ICON-NWP and AFES models is based on an older version of the transport parameterization (using ERA Interim and fewer years for the ATLAS-SWIFT runs). Table 2 shows the fitted parameters for the ECHAM, ICON-NWP and AFES implementa-

tion, which are slightly different.

**Table 1.** Fit coefficients (current version). For a version with error bars, see supplement Table S1

| $p$ [hPa] | 69.66111 | 54.03643 | 41.59872 | 31.77399 | 24.07468 | Unit |
|---|---|---|---|---|---|---|
| Constant change term NH ($c_{\mathrm{const}}$) | 0.0888 | 0.1050 | 0.1068 | 0.0969 | 0.0793 | $\cdot 10^{-7}\mathrm{day}^{-1}$ |
| Constant change term SH ($c_{\mathrm{const}}$) | 0.1338 | 0.4850 | 0.7423 | 0.9217 | 0.9539 | $\cdot 10^{-8}\mathrm{day}^{-1}$ |
| Temperature-dependent term NH ($c_{\mathrm{T}}$) | 0.2814 | 0.2841 | 0.2221 | 0.1489 | 0.0579 | $\cdot 10^{-7}\mathrm{K}^{-1}$ |
| Temperature-dependent term SH ($c_{\mathrm{T}}$) | 0.2533 | 0.3097 | 0.3152 | 0.2775 | 0.1375 | $\cdot 10^{-7}\mathrm{K}^{-1}$ |

**Table 2.** Fit coefficients (used in ECHAM, ICON-NWP and AFES)

| $p$ [hPa] | 69.66111 | 54.03643 | 41.59872 | 31.77399 | 24.07468 | Unit |
|---|---|---|---|---|---|---|
| Constant change term NH ($c_{\mathrm{const}}$) | 0.0751 | 0.0943 | 0.1010 | 0.1004 | 0.0992 | $\cdot 10^{-7}\mathrm{day}^{-1}$ |
| Constant change term SH ($c_{\mathrm{const}}$) | 0.0944 | 0.4633 | 0.6919 | 0.7896 | 0.7704 | $\cdot 10^{-8}\mathrm{day}^{-1}$ |
| Temperature-dependent term NH ($c_{\mathrm{T}}$) | 0.2162 | 0.2277 | 0.1689 | 0.1049 | 0.0135 | $\cdot 10^{-7}\mathrm{K}^{-1}$ |
| Temperature-dependent term SH ($c_{\mathrm{T}}$) | 0.1251 | 0.2423 | 0.2689 | 0.2293 | -0.0204 | $\cdot 10^{-7}\mathrm{K}^{-1}$ |

## 3.1 Fit to the transport of ozone modelled by ATLAS-SWIFT

We use a multivariate regression model to obtain the fit parameter $c_{\mathrm{T}}$ for the temperature-dependent term and the fit parameter $c_{\mathrm{const}}$ for the "constant change" term in each model layer from the ATLAS-SWIFT runs. The regression model is based on a fit to the ozone change per day by transport $\Delta\mathrm{O}_{3,\mathrm{ATLAS,transport}}$ in ATLAS-SWIFT, which we will derive before discussing the regression model.

The vortex-averaged total modelled ozone change per day in a layer $\Delta\mathrm{O}_{3,\mathrm{ATLAS,total}}$ in the ATLAS-SWIFT runs is separated into the ozone change per day by transport $\Delta\mathrm{O}_{3,\mathrm{ATLAS,transport}}$ and the ozone change per day by chemistry $\Delta\mathrm{O}_{3,\mathrm{ATLAS,chem}}$:

$$\Delta\mathrm{O}_{3,\mathrm{ATLAS,total}} = \Delta\mathrm{O}_{3,\mathrm{ATLAS,chem}} + \Delta\mathrm{O}_{3,\mathrm{ATLAS,transport}} \tag{2}$$

All quantities are based on ozone volume mixing ratios. The vortex-averaged ozone change by chemistry per day in a layer $\Delta\mathrm{O}_{3,\mathrm{ATLAS,chem}}$ is obtained by the chemical rate of change of ozone from the ATLAS-SWIFT model. Throughout any given layer in the vortex, the ozone change by chemistry is constant and is a direct result of the equations of the Polar SWIFT model. Next, the total modelled ozone change per day in a layer $\Delta\mathrm{O}_{3,\mathrm{ATLAS,total}}$ is obtained by the difference in the modelled vortex-averaged ozone in a layer between two consecutive days. Finally, the vortex-averaged ozone change by transport per day in a layer $\Delta\mathrm{O}_{3,\mathrm{ATLAS,transport}}$ is the difference between the total modelled change and the change in chemistry.

All time series of the vortex-averaged values are limited to include only values when the vortex existed. For each year and hemisphere, a date of "vortex formation" and a date of "vortex breakup" are defined. There is some uncertainty and arbitrariness in these dates, since these are long-term processes, but the sensitivity of the results to the dates chosen is small

**Table 3.** Assumed dates of vortex formation and breakup in the northern hemisphere

| Year | Formation | Breakup | Year | Formation | Breakup |
|------|-----------|---------|------|-----------|---------|
| 1980 | 29 Dec | 14 Mar | 2001 | 26 Jan | 18 Feb |
| 1981 | 11 Dec | 15 Feb | 2002 | 15 Dec | 16 Feb |
| 1982 | 13 Dec | 26 Mar | 2003 | 10 Dec | 20 Mar |
| 1983 | 3 Dec | 10 Mar | 2004 | 14 Dec | 22 Jan |
| 1984 | 4 Dec | 5 Mar | 2005 | 20 Dec | 24 Mar |
| 1985 | 7 Dec | 18 Jan | 2006 | 2 Jan | 28 Jan |
| 1986 | 28 Dec | 21 Mar | 2007 | 11 Dec | 10 Mar |
| 1987 | 10 Dec | 31 Jan | 2008 | 24 Dec | 1 Mar |
| 1988 | 24 Jan | 23 Mar | 2009 | 18 Dec | 5 Feb |
| 1989 | 12 Dec | 7 Mar | 2010 | 6 Jan | 21 Feb |
| 1990 | 15 Dec | 30 Mar | 2011 | 10 Dec | 30 Mar |
| 1991 | 26 Nov | 2 Feb | 2012 | 2 Dec | 5 Feb |
| 1992 | 7 Dec | 29 Mar | 2013 | 10 Dec | 24 Jan |
| 1993 | 12 Dec | 22 Mar | 2014 | 11 Dec | 30 Mar |
| 1994 | 14 Dec | 31 Mar | 2015 | 13 Dec | 25 Mar |
| 1995 | 6 Dec | 29 Mar | 2016 | 30 Nov | 11 Mar |
| 1996 | 12 Dec | 30 Mar | 2017 | 4 Jan | 27 Feb |
| 1997 | 17 Jan | 30 Mar | 2018 | 19 Dec | 22 Feb |
| 1998 | 17 Dec | 28 Feb | 2019 | 2 Dec | 1 Mar |
| 1999 | 13 Dec | 1 Jan | 2020 | 12 Dec | 30 Mar |
| 2000 | 19 Dec | 18 Mar | 2021 | 9 Dec | 7 Feb |

(changing the dates within $\pm 10$ days does change the fit coefficients by less than 10% typically). For the northern hemisphere, vortex formation is defined as the date when the area of the vortex first exceeds 15 million $km^2$ at 54 hPa. Vortex breakup is defined as the last date when the area of the vortex was above 15 million $km^2$ at 54 hPa (but limited to the last date of the model run when the vortex existed longer than 31 March). These values can be found in Table 3. For the southern hemisphere, a "formation" date of 15 May and a "breakup" date of 31 October are assumed for all years.

As an example, Figure 1 shows the total change (a, b), the change by chemistry (c, d) and the change by transport (e, f) obtained in this way for the northern hemispheric winter 2010/2011 and the southern hemispheric winter 2011 at 54 hPa (layer 2) as a function of the day of year. Figures for all years, layers and both hemispheres can be found in a supplement (Figures S1–S15 and S26–S40). For clarity and to reduce noise, we show the changes cumulated over time in Figure 1. However, the following fit is based on the non-cumulated changes from Equation 2. All cumulated time series in Figure 1 start with 0 at the vortex formation date.

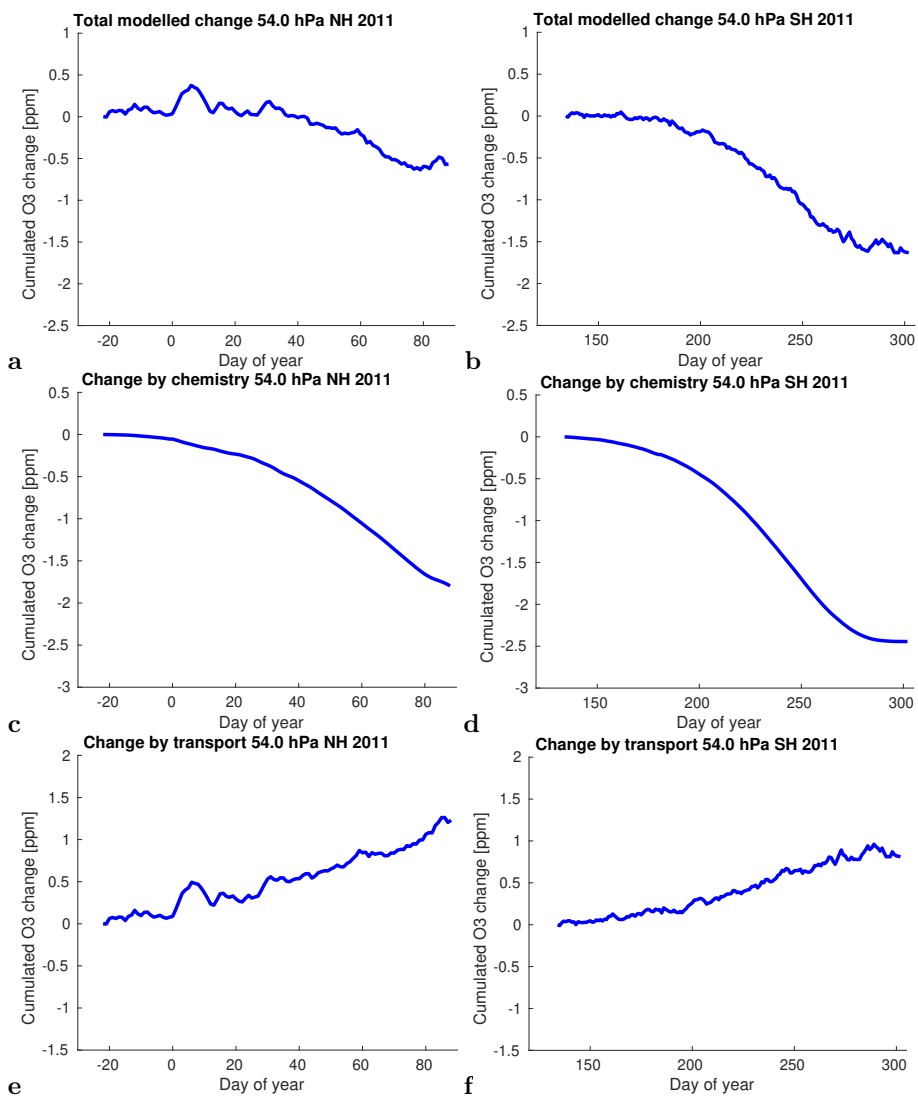

**Figure 1.** Cumulated change of vortex-averaged ozone mixing ratio for the northern hemispheric winter 2010/2011 (a, c, e) and the southern hemispheric winter 2011 (b, d, f) at 54 hPa simulated by ATLAS-SWIFT as a function of the day of year. Panels (a, b) show the total simulated change, panels (c, d) show the change by chemistry and panels (e, f) show the change by transport.

We use the following multivariate regression model to obtain the fit parameter $c_T$ for the temperature-dependent term and the fit parameter $c_{const}$ for the "constant change" term in each model layer:

$$\Delta \hat{O}_{3,\text{ATLAS,transport}}(t) = c_T(\Delta T_{\text{ERA5}}(t) - \Delta T_R'(t)) + c_{\text{const}}' \tag{3}$$

$\Delta \hat{O}_{3,\text{ATLAS,transport}}(t)$ is a least-squares fit to the time series of the ozone change by transport per day $\Delta O_{3,\text{ATLAS,transport}}(t)$ in the layer as defined in Equation 2. For the fit, we concatenate the time series of all individual years between the start dates

and the end dates (Table 3) to a single time series. $\Delta T_{\text{ERA5}}(t)$ is the corresponding change of vortex-averaged temperature per day obtained from ECMWF ERA5 data (don't confuse with $\Delta T_{\text{GCM}}(t)$), and $\Delta T_R'(t)$ is the change of the vortex-averaged radiative equilibrium temperature per day (don't confuse with $\Delta T_R(t)$ from Equation 1, which is the change within the time step of the GCM). $\Delta T_R'(t)$ is based on a fixed table of values and only depends on the day of year (see Section 3.3 and source code in the Zenodo repository). $c_{\text{const}}'$ is related to $c_{\text{const}}$ from Equation 1 by $c_{\text{const}}' = c_{\text{const}}\Delta t'$, where $\Delta t'$ is the time period

of one day used to obtain $\Delta O_{3,\text{ATLAS,transport}}(t)$, $\Delta T_{\text{ERA5}}(t)$ and $\Delta T_R'(t)$.

     As an example, Figure 2 (a) shows the concatenated time series of the change of vortex-averaged ozone mixing ratio by transport per day in ATLAS-SWIFT at 54 hPa (layer 2) in the northern hemisphere for all years (blue) and the corresponding fit from the regression model (red). Figure 2 (b) shows the same for the southern hemisphere. Figure 2 (c, d) show the same data as a scatter plot for clarity. Corresponding figures for other levels can be found in the supplement (Figure S16 and S41).

A discussion of the rationale behind choosing this regression model is now given in the following in Sections 3.2 and 3.3, including the derivation of an estimate for the radiative equilibrium temperature.

## 3.2    Rationale behind the "constant change" term

Figure 1 (e, f) shows that the cumulated vortex-averaged ozone change by transport at 54 hPa (layer 2) for the northern hemispheric winter 2010/2011 and the southern hemispheric winter 2011 as a function of time is linear to a good approximation

(meaning that the non-cumulated change is constant to good approximation). It turns out that this also is a good approximation for all other years and layers (see Figures S11–S15 and S36–S40 in the supplement). This is also confirmed later by the validation results: Figure 5 shows the same cumulated vortex-averaged ozone change as in Figure 1 (blue) and the fitted "constant change" term (grey), see Figures S19–S23 and S44–S48 in the supplement for other years and layers. This suggests an approach where a constant amount of ozone is added in each layer per time step of the SWIFT model.

## 3.3    Rationale behind the temperature-dependent term

The temperature-dependent term adds variability to the transport parameterization. Variability in ozone transport is caused by variability in the Brewer-Dobson circulation (e.g. Andrews et al., 1987; Fusco and Salby, 1999; Randel et al., 2002; Weber et al., 2003). Vortex temperatures, the descent in the vortex, and the transport over the vortex edge are all related by the mechanisms of the Brewer-Dobson circulation. A stronger Brewer-Dobson circulation is related to more adiabatic heating and

higher temperatures in the vortex, which cause more diabatic cooling and descent in the polar vortex. However, note that there

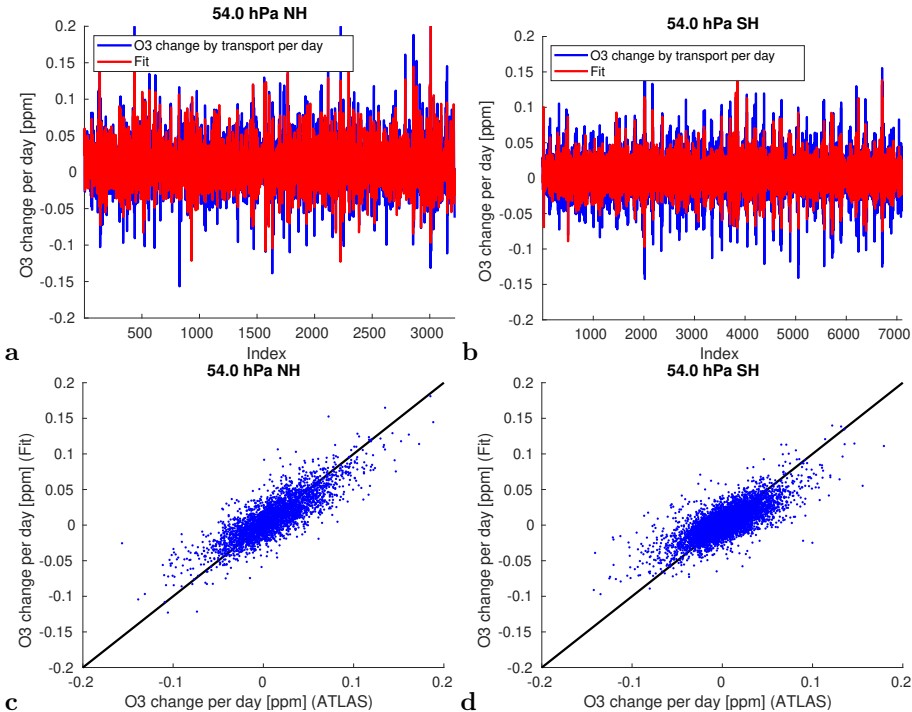

**Figure 2.** Concatenated time series of the change of vortex-averaged ozone mixing ratio by transport per day for all years in ATLAS-SWIFT at 54 hPa (blue) and corresponding fit from the multivariate regression model (red). (a) Northern hemisphere, (b) Southern hemisphere. (c) and (d) show the same data as a scatter plot.

may be temperature variations on a pressure level not caused by the Brewer-Dobson circulation, as changes caused by low and high pressure systems from the troposphere.

When assuming a temporally constant zonally averaged radiative equilibrium temperature $\overline{T}_{\mathrm{R}}$, the relationship between zonal mean temperature at a given date $\overline{T}(0)$ and the downwelling $\overline{w}^*$ (vertical residual velocity in a zonal mean sense in log-pressure coordinates) of the Brewer-Dobson circulation is approximated by

$$\overline{T}(0) = \overline{T}_{\mathrm{R}} + \exp(-\alpha t)(\overline{T}(-t) - \overline{T}_{\mathrm{R}}) - \int_{-t}^{0} \overline{w}^* S \exp(\alpha t')\,\mathrm{d}t' \tag{4}$$

where $1/\alpha$ is the radiative relaxation time scale (about 1 month), and $S$ is static stability (i.e., the vertical gradient of potential temperature) (see e.g. Andrews et al., 1987). That is, the temperature difference from the radiative equilibrium temperature is basically the integral over the downwelling in the past, weighted by an e-folding time of about 1 month. On short time scales, the change in temperature is directly correlated to the corresponding downwelling in that time period ($\overline{T}_{\mathrm{R}}$ terms nearly cancel out, since $\exp(-\alpha t) \approx 1$). As a complication, it has to be considered that $\overline{w}^*$ has to be multiplied by the vertical gradient $\mathrm{d}\chi/\mathrm{d}z$ of a species $\chi$ (as ozone) to obtain the change by transport in this species. This vertical gradient can vary over winter.

As mentioned, the temperature-dependent term is based on an input variable that is readily available from the GCM, which is the change in vortex-averaged temperature in a layer per time step of the model $\Delta T_{\mathrm{GCM}}$. The corresponding variable in the regression model is the vortex-averaged temperature change $\Delta T_{\mathrm{ERA5}}$ per day from ERA5 in a layer. We now try to find a well-working empirical relationship between $\Delta T_{\mathrm{ERA5}}$ and the change in ozone by transport per day which is roughly based on the physical considerations explained above. However, we deliberately choose to simplify our approach as much as possible.

Equation 4 suggests that the change in temperature per day roughly corresponds to the strength of the downwelling during this day caused by the Brewer-Dobson circulation. In turn, the magnitude of downwelling can be assumed to be correlated to the change of ozone per day by transport of ozone-rich air from above.

Equation 4 assumes a constant radiative equilibrium temperature. However, the change in temperature from day to day is not only caused by the Brewer-Dobson circulation, but also by the change of the radiative equilibrium temperature with time. For this reason, we subtract an approximation of the change of the vortex-averaged radiative equilibrium temperature in the transport parameterization (Equation 1) and the multivariate regression (Equation 3). The radiative equilibrium temperature is assumed to be identical on the same calendar day for any individual year. As a rough approximation for the radiative equilibrium temperature, we simply take the average of the vortex-averaged temperature over all years minus a constant offset. Figure 3 shows the vortex-averaged temperatures for 54 hPa in the northern hemisphere (a) and southern hemisphere (b) for all years (grey lines), and the average over all years (red line). This figure gives us confidence that our approach is feasible: A version of the red line for the average over all years that is shifted down by an constant offset of a few Kelvin (black line) corresponds roughly to the lower envelope of all the grey lines. Assuming that the lower envelope shows situations with a weak Brewer-Dobson circulation where the vortex-averaged temperature approaches the radiative equilibrium temperature, the black curve can be seen as an approximation of the radiative equilibrium temperature. The constant offset of the black line to the red line plays no role for the results, since it cancels out when calculating the change of the radiative equilibrium temperature per day. Note that the approximation of the radiative equilibrium temperature obtained in this way shows the expected temporal evolution, i.e., lower temperatures during polar night.

Figure 4 shows that the change of vortex-averaged temperature per day at 54 hPa from ERA5 for all days in all years is well correlated to the corresponding change of vortex-averaged ozone mixing ratio per day in ATLAS-SWIFT, when the change in radiative equilibrium temperature is subtracted. The correlation coefficient at 54 hPa is 0.84 for the northern hemisphere and 0.75 for the southern hemisphere. The good correlation between the ozone change and the temperature change shows that the day-to-day variations of ozone are well represented by the temperature-dependent term of the parameterization, despite the simplifications in the process of obtaining this empirical relation. An effect of the constant term of the parameterization, which would result in a vertical shift of the point cloud away from the origin, cannot be observed in Figure 4, because this term is significantly smaller than the temperature-dependent term (see Table 1). Nevertheless, the constant term dominates the total ozone change over the course of the winter, because it acts persistently, whereas the effects of the temperature changes cancel to a large extent (cf. Figure 5).

Figure 4 also shows that the correlation is similar in the northern and southern hemisphere. This can also be seen by the similar values of the fit parameters for the temperature-dependent term in the southern and northern hemisphere in Table 1.

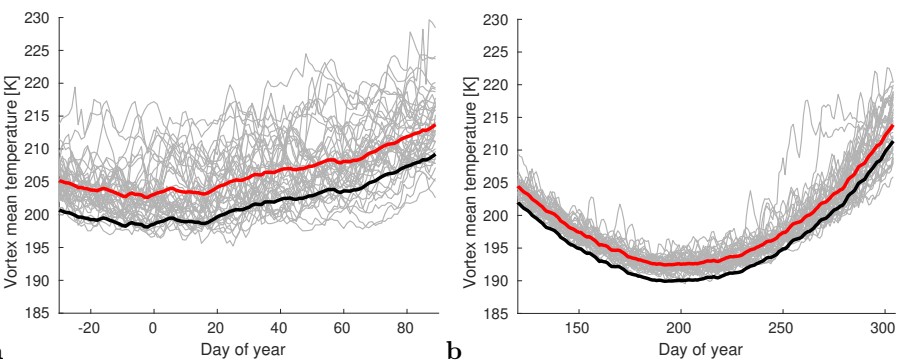

**Figure 3.** Vortex mean temperatures at 54 hPa as a function of day of year for all individual years from 1979/1980 to 2020/2021 (northern hemisphere, a) or 1980–2021 (southern hemisphere, b) based on ERA5 (grey lines), vortex mean temperature averaged over all years (red) and the same curve shifted by $-4.5\,\mathrm{K}$ or $-2.5\,\mathrm{K}$ as an approximation of the lower envelope of the grey lines (black).

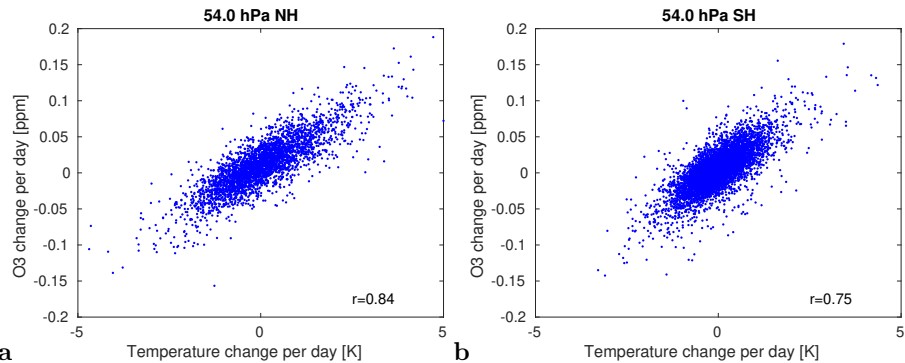

**Figure 4.** Scatter plot of the change of vortex-averaged ozone mixing ratio by transport per day at 54 hPa for all years in ATLAS-SWIFT and corresponding change in vortex-averaged temperature per day from ERA5, corrected for the change in radiative equilibrium temperature. (a) Northerm hemisphere, (b) Southern hemisphere.

As a note of caution we have to stress here that while this method will work well for short-term changes in temperature and ozone, it might not work well for changes on a longer time scale. On longer time scales, temperature will start to lose memory

of the transport in the past due to the radiative relaxation time scale of about 1 month.

## 4   Validation

We use two different methods for validation. First, we validate only the transport term by running a stand-alone version of the transport parameterization that uses ERA5 data. This stand-alone version is not implemented into a GCM and has no chemistry. Then, we validate a stand-alone version of the complete Polar SWIFT model (chemistry plus transport parameterization),

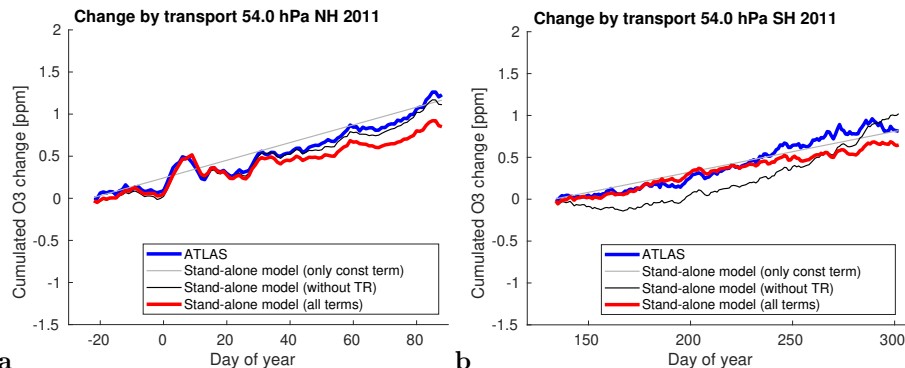

**Figure 5.** Cumulated vortex-averaged ozone change by transport at 54 hPa for the northern hemispheric winter 2010/2011 (a) and the southern hemispheric winter 2011 (b) as in Figure 1 (e,f) (blue) and a simulation of the cumulated change by transport by a stand-alone version of the transport parameterization (red). The thin grey line shows a simulation with only the "constant change" term, and the thin black line shows a simulation with the "constant change" and temperature-dependent term, but without subtracting the change of the radiative equilibrium temperature from the vortex-averaged temperature change.

again driven by ERA5 data. This version is again not implemented into a GCM, but will give the same results as a version implemented into a GCM that would simulate the same temperatures and polar vortex as ERA5.

### 4.1  Stand-alone version of the transport parameterization

First, we calculate the cumulated vortex-averaged ozone change by transport by directly applying Equation 1. Vortex-averaged temperatures are taken from ERA5. The transport parametrization is run only for the days when a minimum size of the vortex

of 15 million $km^2$ at 54 hPa is exceeded, in accordance with the time periods used for the fit. Starting values are set to zero. As an example, Figure 5 shows the simulated cumulated change by transport at 54 hPa (layer 2) for the northern hemispheric winter 2010/2011 and the southern hemispheric winter 2011 (red), compared to the original transport term from the ATLAS-SWIFT run (blue). To show the contribution of the different terms of Equation 1, the thin grey line shows a simulation with only the "constant change" term, and the thin black line shows a simulation with the "constant change" and temperature-dependent

term, but without subtracting the change of the radiative equilibrium temperature from the vortex-averaged temperature change. See Figures S19–S23 and S44–S48 in the supplement for other years and layers.

The difference between the cumulated vortex-averaged ozone change by transport simulated by ATLAS-SWIFT and simulated by the transport parameterization at the date of vortex breakup is typically about 0.2 ppm in the northern hemisphere (see Figure S25 of the supplement). This is in the order of magnitude of 10 % of the simulated or observed ozone at these dates

(see next section 4.2 and Figure S25). The differences in the southern hemisphere are somewhat larger and can reach more than 0.5 ppm and in the order of magnitude of 20 % of the simulated or observed ozone at these dates (see Figure S50 of the supplement).

Figure 5 shows that while the parameterization is able to capture the short-term variations in ozone quite well, and also captures the general increase in the cumulated ozone change by transport with time, it has difficulties to get the exact magnitude of the increase right in 2010/2011. Figures S19–S23 and S44–S48 from the supplement show that the magnitude of increase is met quite well in many other years, but that sometimes, it is underestimated or overestimated, with no clear pattern. These difficulties to model the long-term change correctly may be related to the temperature-dependent term and the parameterization might not work well for time periods longer than the radiative relaxation time scale.

## 4.2 Complete stand-alone version of SWIFT

The transport parameterization is also validated by runs of a stand-alone version of the complete Polar SWIFT model. The stand-alone version calculates the vortex-averaged ozone mixing ratios as a function of time for the levels that SWIFT is based on. The model is initialized with values for ozone, HCl, $HNO_3$ and $ClONO_2$ on the start date. Initial values are taken from the MLS climatologies (i.e., they are the same for every year). The change of ozone per time step is calculated as the sum of the change by chemistry from the Polar SWIFT model and the change by transport from the transport parameterization. The stand-alone model is driven by meteorological data from ERA5 again. Runs start on 1 November for the northern hemisphere and on 1 May for the southern hemisphere. The transport parametrization is only switched on when a minimum size of the vortex of 15 million $km^2$ at 54 hPa is exceeded (mainly to account for the fact that the date of vortex formation is usually later than the start date of the model run). Chlorine is scaled by EESC in the same way as described above for the ATLAS-SWIFT model. Simulations of the Arctic winters 1979/1980–2020/2021 and the Antarctic winters 1980–2021 are performed. Results are compared to actual measurements of the MLS instrument on a given date for validation.

Figure 6 a shows the simulated vortex-averaged ozone at 54 hPa (layer 2) for the date of vortex breakup (see Table 3, but note that in 2011, we use 25 March in the plot due to an instrument failure of the MLS instrument) in the northern hemisphere for different years (see Figure S24 in the supplement for other layers). The blue line shows the ozone mixing ratios simulated without the transport parameterization (i.e., only the chemistry of the Polar SWIFT model), the brown line shows the ozone mixing ratios simulated with only the "constant change" term of the transport parameterization, and the green line shows the ozone mixing ratios simulated with the full transport parameterization with the "constant change" term and temperature-dependent term. The red line shows corresponding measurements of ozone from the MLS instrument. The rationale behind showing the simulation results at the day of vortex breakup is to have a time period as long as possible in each individual winter where SWIFT is able to simulate the changes in ozone, so that we get a worst-case estimate of potential systematic errors that add up during the model run.

Both runs with the transport parameterization agree considerably better with the measurements than the run with only chemistry. However, the difference between the full transport parameterization with the temperature-dependent term and the parameterization using only the "constant change" term is relatively small. Actually, while the temperature-dependent term captures the short-term changes in ozone quite well (see Figure 5, and Figures S19–23, S44–48), it does not significantly improve the simulation of the long-term changes in ozone. To repeat this again, this may be related to the fact that the temperature-dependent term might not work well for time periods longer than the radiative relaxation time scale. In fact, the transport parameterization

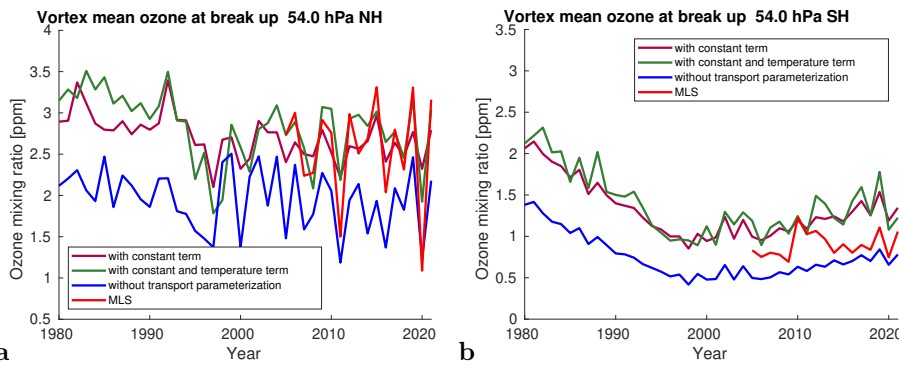

**Figure 6.** Vortex-averaged ozone at 54 hPa simulated by the stand-alone Polar SWIFT model for the date of vortex breakup (see Table 3, but note that in 2011, we use 25 March in the plot due to an instrument failure of the MLS instrument) in the northern hemisphere (a) and southern hemisphere (b) for different years. Ozone mixing ratios simulated without the transport parameterization (blue), ozone mixing ratios simulated with only the "constant change" term of the transport parameterization (brown), ozone mixing ratios simulated with the full transport parameterization with the "constant change" term and temperature-dependent term (green), and corresponding measurements of ozone from the MLS instrument (red).

without the temperature-dependent term does not perform worse than the parameterization with the temperature-dependent term on average for the ozone values at the vortex breakup date (Figures 6, 7, S24, S49).

We have calculated the root mean square error (RMSE) of the difference of the simulated ozone of the stand-alone models at the time of vortex breakup to the MLS measurements and the correlation coefficient of the same quantities to give a more quantitative account of this. For instance, the RMSE of the parameterization with the temperature-dependent term at 54 hPa in the northern hemisphere is 0.36 ppm, while it is 0.44 ppm for the parameterization with only the "constant change" term. The correlation coefficient is 0.88 and 0.86, respectively. Values for the RMSE and the correlation coefficients can be found in Figure 7 and in the Figures S24 and S49 in the supplement for the other levels and the transport parameterization without the

temperature-dependent term. It is apparent that sometimes the parameterization with the temperature-dependent term performs better in terms of RMSE, and sometimes the parameterization without the term does perform better, and that the same is true for the correlation coefficient, with no clear pattern. This indicates that there are opportunities for improvement of the parameterization.

     Next, we compare the model results with the measurements as a function of the magnitude of the ozone depletion and of

the magnitude of the transport in the winter. Figure 7 a shows a scatter plot of the modelled ozone (with the full transport parameterization) versus the observed ozone from MLS at 54 hPa (based on the same data as in Figure 6), see Figure S24 in the supplement for other layers and the transport parameterization without the temperature-dependent term. The interannual variability is usually captured quite well in the northern hemisphere at the first 4 levels, which can be seen from the correlation coefficients of 0.75–0.88 at these levels. RMSE values of 0.21–0.36 ppm at levels 2–4 are also reasonable values in comparison

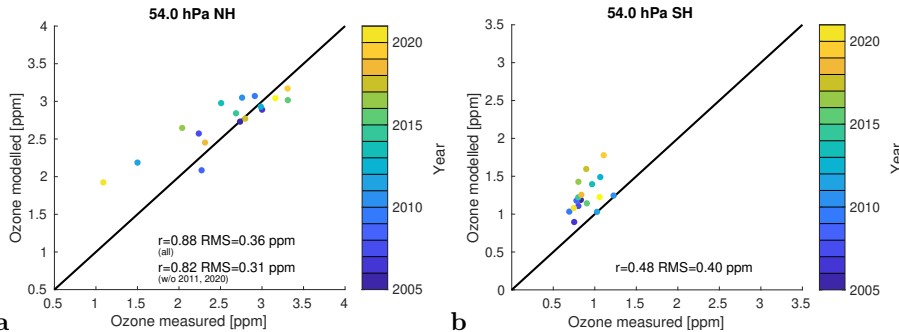

**Figure 7.** Scatter plots of the modelled ozone at the date of vortex breakup (with the full transport parameterization) versus the observed ozone from MLS (based on the same data as in Figure 6).

to the observed and simulated ozone values of 2–3 ppm. However, there is a noticeable bias at the first level (69 hPa), which leads to a RMSE of 0.67 ppm.

Note that the simulated ozone in the figures does not only depend on the transport parameterization, but also in large parts on the chemistry simulation of Polar SWIFT and on systematic differences in the vortex breakup dates, which determine for how long the transport parameterization is actually applied. E.g., the vortex breakup is later on average in cold winters with a large ozone depletion. Hence, we will not go into further detail here in interpreting the differences, since this would lead to far from the focus of this paper and is better suited to a validation study of the complete SWIFT model.

The model overestimates ozone at 54 hPa by about 0.7–0.8 ppm compared to the MLS measurements in two winters with low ozone values (2010/2011 and 2019/2020). That is, in cold winters with large ozone depletion and a weak Brewer-Dobson circulation, while warmer years are simulated relatively well (however, the overestimation in cold winters is much less pronounced at the levels 3–5, see Figure S24 in the supplement). The differences in cold winters between Polar SWIFT and MLS at 54 hPa might not be explained by the transport parameterization alone, since they are much larger than the differences of about 0.2 ppm between the transport parameterization and the transport term of ATLAS discussed in Section 4.1 (however, this relies on the assumption that the transport is represented well in the ECMWF ERA5 data and in the transport scheme of ATLAS). Hence, the differences between Polar SWIFT and MLS in cold winters could also be a deficiency of the chemistry model of Polar SWIFT. However, there is no clear indication from Wohltmann et al. (2017) that this could be the case.

Figure 6 b and Figure 7 b show the same for the southern hemisphere (see Figure S49 in the supplement for other layers). While interannual variability is lower in the southern hemisphere, the model has difficulties to get the mean ozone values correct at some levels compared to the MLS measurements. That may, e.g., be related to the fact that the transport derived from the meteorological data of ECMWF in the southern hemisphere is not represented well or that there are deficiencies in the Polar SWIFT chemistry model. Figure 16 of Wohltmann et al. (2017) shows an ozone bias at 46 hPa for the mean ozone values for a Polar SWIFT run with the full transport scheme of ATLAS that is similar to the bias observed at 41.6 hPa for run with the transport parameterization (Figure S49 of the supplement). This points into the direction that the transport parameterization is not a likely cause for the differences.

## 5 Conclusions

We present a transport parameterization for the Polar SWIFT model that simulates the change of vortex-averaged ozone by transport in a fast and simple way without the need for a complex transport scheme in the GCM. A benefit of this approach includes the easy and self-contained coupling to a GCM. Another advantage can be that a transport parameterization based on reanalysis data and measurements can avoid deficiencies in the representation of transport in the GCMs. At the moment, the transport parameterization and Polar SWIFT are implemented in the ECHAM6, ICON-NWP and AFES4.1 GCMs as an

alternative to the transport options in these models. We derived the equations for the transport parameterization, fitted the parameterization to the transport simulated in the ATLAS model and validated the parameterization by simulating the original transport term from ATLAS and by comparing the complete Polar SWIFT model to MLS satellite observations. The results of the transport parametrization agree well with the results of the detailed transport scheme of ATLAS, with a typical difference of 0.2 ppm in the simulated cumulated change of ozone volume mixing ratio by transport at the time of polar vortex break-up in the

northern hemisphere and 0.5 ppm in the southern hemisphere. This is about 10%–20% of the obserevd ozone at this date. The constant term of the of the transport parametrization generates a significantly larger contribution to the change of polar ozone over the course of winter than the temperature term. Agreement of the complete model (including chemistry) with observations is usually better in the northern hemisphere than in the southern hemisphere (see Figure 6, Figure 7 and supplement). For instance, the root mean square error compared to MLS observations of vortex-averaged ozone at vortex breakup at 54 hPa in

the northern hemisphere is 0.36 ppm, and the correlation to MLS is 0.88, while in the southern hemisphere RMSE is 0.4 ppm and the correlation is 0.48.

*Code and data availability.* The code repository for the ATLAS model can be assessed at https://gitlab.awi.de/iwohltmann/atlas. The scripts used for this manuscript, including the SWIFT stand-alone model, the stand-alone model of the transport parameterization and the start script for ATLAS-SWIFT, are available at doi:10.5281/zenodo.5834222 at Zenodo. MLS data are available at https://disc.gsfc.nasa.gov/datasets?page=1&keywor

ECMWF ERA5 data are available at https://cds.climate.copernicus.eu/cdsapp#!/home.

*Author contributions.* IW developed the transport parameterization, performed the validation runs, wrote the manuscript and produced the figures. DK implemented SWIFT in the ECHAM, AFES and ICON-NWP models and contributed to the discussion of GCMs. RL contributed to the development of the transport paramerization. All authors discussed the results and developed the manuscript.

*Competing interests.* The authors declare that they have no conflict of interest.

*Acknowledgements.* We thank the MLS Science Team for producing and making public the MLS data. We thank ECMWF for providing ERA5 reanalysis data, generated using Copernicus Climate Change Service Information 2020. Neither the European Commission nor ECMWF is responsible for any use that may be made of the Copernicus Information or Data it contains. Copernicus Climate Change Service (C3S) (2017): ERA5: Fifth generation of ECMWF atmospheric reanalyses of the global climate. Copernicus Climate Change Service Climate Data Store (CDS), 2017–2020.

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
