# Peer review of "Transport parameterization of the Polar SWIFT model (version 2)"

_Geoscientific Model Development, 2022_

## Author Comment (AC1)

Dear Slimane Bekki,

here is the missing reply to your comment from the pre-review:

- *Also, different parameters of the scheme are derived from fits to the AT-LAS modelled ozone evolution for different years and then to the averaging of the fit coefficients for the 40 years. The tables of the fit coefficients show single values without any error bars/standard deviations. You should provide values with uncertainties in the manuscript.*

  To be able to resolve this comment, we had to change the method of obtaining the fit coefficients. Unfortunately, it was mathematically not obvious how to obtain these values with the method used in the original manuscript.

  For this reason, we introduced 2 major changes to the method:

  1. The fit is now based on the non-accumulated ozone time series (i.e., changes per day) and not on the accumulated time series.

  2. Instead of fitting the parameters for each individual year and then averaging the fit parameters over the years, we now concatenate the time series of all years before the fit to obtain a single fit parameter.

  95% confidence intervals for the single fit parameters are obtained by the method presented in Wohltmann et al., 2007, doi:10.1029/2006JD007573, paragraph 41. While this method considers auto-correlation in the residuals, it turned out that the change from accumulated to non-accumulated time series actually led to very low auto-correlations, and this would not have been necessary (but does not hurt either).

  These changes prompted the following changes to the manuscript:

  1. A complete rewrite of section 3 to reflect the changes in the method.

  2. Addition of new Figures 2 and 4 (figure numbers from new manuscript) and deletion of old Figures 2 and 3 (figure numbers from original manuscript).

  3. The content of all figures and the numbers given in the text (fit parameters etc.) have changed sligthly throughout the paper.

  4. We have now added a version of the table with error bars in the supplement and have added a reference to this table in the table caption.

  The fit parameters obtained by the new method are very similar to the fit parameters of the old method. That means that the results do not change qualitatively and that the conclusions remain the same.

  The new method is more elegant and gets rid of some (partly arbitrary) assumptions of the old method (see also comment of reviewer 1 to lines

132–133 that is resolved by the new method). However, a disadvantage is that the actual fit does not "look" as clear and intuitive in new Figure 2 as this was the case with old Figures 2 and 3.

---

## Author Comment (AC2)

Dear reviewer,
many thanks for your helpful comments!

**Important note**

There is a major change unrelated to the reviewer comments: To be able to resolve a comment of the editor, we had to change the method of obtaining the fit coefficients. The editor asked for error bars in Table 2 (Table 1 in the revised manuscript). Unfortunately, it was mathematically not obvious how to obtain these values with the method used in the original manuscript. For this reason, we introduced 2 major changes to the method:

1. The fit is now based on the non-accumulated ozone time series (i.e., changes per day) and not on the accumulated time series.

2. Instead of fitting the parameters for each individual year and then averaging the fit parameters over the years, we now concatenate the time series of all years before the fit to obtain a single fit parameter.

These changes prompted the following changes to the manuscript:

1. A complete rewrite of section 3 to reflect the changes in the method.

2. Addition of new Figures 2 and 4 (figure numbers from new manuscript) and deletion of old Figures 2 and 3 (figure numbers from original manuscript).

3. The contents of all figures and the numbers given in the text (fit parameters etc.) have changed sligthly throughout the paper.

The fit parameters obtained by the new method are very similar to the fit parameters of the old method. That means that the results do not change qualitatively and that the conclusions remain the same.

The new method is more elegant and gets rid of some (partly arbitrary) assumptions of the old method (see also comment to lines 132–133 that is resolved by the new method). However, a disadvantage is that the actual fit does not "look" as clear and intuitive in new Figure 2 as this was the case with old Figures 2 and 3.

**Note on dates for vortex formation and breakup**

During the preparation of the revised version, we noticed some slight inconsistencies in the definition of the vortex formation and breakup dates. A few dates were not consistent with the 15 million $km^2$ criterion (see "tracked changes" version of Table 3), and the validation in Figure 6 and 7 of the revised manuscript did not use exactly the same dates as the fit. This has been corrected. In addition, we changed the vortex formation date for the southern hemisphere from 1 May to 15 May to exclude some time periods with a weak vortex.

**Note on plots**

For technical reasons, we had to change the software used to create the plots. That means that colors, font sizes, axis tick marks etc. may have changed.

**General comments**

- *Although the description of the previous implementation of Polar SWIFT, the one requiring additional tracers advected by the model, and how this differs from the revised version would be helpful.*

  There is no previous implementation of SWIFT into a GCM that does not use the transport parameterization, and hence, this is not a revised version, but the original implementation.

  The confusion may arise since it was not discussed in Wohltmann et al. (2017) how the the transport would be handled in an actual implementation in a GCM. This was an obvious omission, and part of the motivation of this manuscript was to rectify this omission. At the time of writing of Wohltmann et al. (2017), this omission was deliberate, since we were still in the process of implementing SWIFT into the GCMs at that time.

  We have now added text to the introduction describing how SWIFT is implemented into a GCM (see reply to your next comment). This may also help with this comment.

  We were required by the journal to put a version number into the title. Since the transport parameterization has not really a version number, we chose the same version number used for Polar SWIFT in Wohltmann et al. (2017), since the transport parameterization has always been part of the complete Polar SWIFT model. We hope that has not caused too much confusion, but we are not allowed to remove this number from the title.

  It may also have caused confusion that it was not mentioned that $ClONO_2$, HCl and $HNO_3$ are not included in the transport parameterization. Tests with a transport parameterization for these species showed that the changes induced for $O_3$ were small. Only $O_3$ is returned to the GCM as input for the radiation module. We have added discussion along these lines to the manuscript.

  We have change "evolution" to "chemical evolution" in line 19 to remove a potential source of confusion. Similar changes have been applied throughout the manuscript.

- *In particular, while there is a discussion of how Polar-SWIFT is implemented in ATLAS (Section 2.2), it is difficult to get an idea of how exactly Polar-SWIFT with the newly developed transport parameterization is implemented in a GCM.*

  We have added additional discussion to the introduction to give a general idea about the parts of the implementation into a GCM that are common to all GCMs. However, we would like to keep this discussion short.

A discussion of the implementation into the GCMs was actually not our main aim. Since the implementation of SWIFT is different for every GCM, we felt it would be out of the scope of this manuscript to describe the implementation in detail here. We would like to refer to the individual papers which describe the implementation of SWIFT into the respective GCMs. Unfortunately, as stated in the abstract, the manuscripts for ECHAM, AFES and ICON-NWP are still in preparation, and the existing publication for ECHAM (Romanowsky et al.) does not contain a very detailed account on the model implementation.

A detailed dicussion will also involve the chemistry part of Polar SWIFT, which is why we think a more thorough discussion leads too far away from the focus of this paper (a paper with a scope similar to Wohltmann et al., 2017, might be more appropriate here).

- *What prognostic variables does the model require...*

  If we understand the comment correctly, you mainly refer to the chemical part of the model described in Wohltmann et al. (2017) here. Since the chemical part is described in detail in Wohltmann et al. (2017), and this paper focuses on the transport part, we would like to leave the short description in lines 20–22 as is. This description is sufficient to understand the following and more detail feels out of the scope of this manuscript.

  As stated above, it may have caused confusion that $ClONO_2$, $HCl$ and $HNO_3$ are not included in the transport parameterization. We now state that in the manuscript.

  *...and how does this differ from the previous version that required the explicit calculation of transport.*

  There is no previous version, see above.

- *One other, overarching question I am left with is how does the GCM specify the concentration of ozone outside of the polar vortex when using this version of the Polar SWIFT ozone parameterization? This should be described here.*

  Outside of the polar vortex, the values of the internal ozone climatology of the GCM, which can vary with season, are used as input for the radiation module. Tracers are not advected outside the polar vortex. There is no interpolation applied between the two domains, since the edge of the polar vortex often forms a strong barrier between air masses and strong gradients in species concentrations are common.

  We have added discussion along these lines in an additional paragraph on the GCM implementation to the introduction.

**Specific comments**

- *Lines 4–5: "Many GCMs do not include a usable general scheme for the transport and mixing of chemical species in the stratosphere." As many*

*GCMs and Earth System Models now contain a prognostic treatment of aerosols, I would think a serviceable transport scheme would be more generally available. Is it perhaps more true that many GCMs do not specify a high-enough model lid and a sufficient number of model levels in the stratosphere to adequately resolve the dynamics of the stratosphere?*

We have changed the statements in the abstract to reflect our motivation in a better way. The original formulation may have been somewhat misleading. It is perhaps better to say that our scheme can be used as an alternative to the schemes for tracer transport and mixing that usually exist in GCMs, and that our scheme may have benefits.

The initial idea of Polar SWIFT was to develop a fast and self-contained module to determine polar ozone depletion, with the aim of an easy and straightforward coupling of this module to a GCM. The concept of parameterizing the transport was our first approach, because it kept the technical interface between SWIFT and the GCM very simple. We have added this as an additional motivation to the abstract.

There may be better methods of simulating the transport of stratospheric ozone than our parameterization and these methods are successfully used in existing models (see e.g. the models and validation in Dietmüller et al, doi:10.5194/acp-18-6699-2018). However, since our transport parameterization is fitted to reanalysis data based on measurements, it may actually perform better than the transport scheme in an existing GCM, which may e.g. suffer from deficiencies in the gravity wave parameterization that influence the Brewer-Dobson circulation in the model. This was one motivation for our parameterization that we state now more clearly in the abstract. For instance, we implemented tracer transport for SWIFT in ECHAM6. ECHAM6 (and also the AFES GCM) is a hydrostatic model and the tracer transport is based on a Lin-Rood scheme (Giorgetta et al., 2014, https://mpimet.mpg.de /fileadmin /publikationen /Reports /WEB_BzE_135.pdf). We tested Polar SWIFT in ECHAM6 with tracer transport and found that the tracer transport of ECHAM6 overestimated the ozone concentrations inside the vortex, especially in the southern polar vortex. The results obtained by the transport parameterization actually were an improvement over the version with tracer transport. A reason for the bad performance of the tracer transport may be the overestimation of horizontal transport, which is a known issue in ECHAM6 (Stevens et al. 2013, doi:10.1002/jame.20015). A GCM with a more advanced tracer transport scheme (e.g. ICON) and improved vertical wave propagation will certainly compensate for some of these deficiencies.

The computational cost of adding more tracers to the GCM was not a serious issue. While the running time increased somewhat, this was not the main bottleneck in the computation.

Temperature biases in the GCM might influence the transport parameterization via the temperature dependent term. This issue and its solution are

addressed in the SWIFT coupling paper, which is currently in preparation.

We do not expect that the model lid poses an issue. The GCMs that Polar SWIFT was coupled to have a model top at 1 Pa, which covers the domain of the Brewer-Dobson circulation.

*Isn't this point also a little beside the point because the parameterization being presented here is for the vortex averaged transport effect and would not be the kind of quantity readily calculated by a 3-D advection scheme in a GCM?*

As can be seen by the implementation of SWIFT in the ATLAS-SWIFT model, there is a straightforward way to implement full 3D transport into a GCM that uses the chemistry part of Polar SWIFT, although SWIFT only calculates vortex averages (see description in lines 48–52 of the original manuscript and Wohltmann et al., 2017). But maybe this is not the point you are aiming at?

The transport parameterization presented here may only be of benefit when a full 3D transport scheme is not available or when there are deficiencies in the representation of transport in the full 3D scheme of the GCM. 3D transport from the GCM does not necessarily perform better than the parameterization, since transport from the GCM could have deficiencies that are not present in the transport parameterization, which is based on reanalysis data. There may be cases where a parameterization of the vortex-averaged transport effect may only be a poor replacement for a full 3D transport scheme.

- *Lines 23–34: The section beginning "When using the Polar SWIFT model. . . " is actually describing the present work but it is a bit disorienting to the reader because it appears in the introduction and really does not describe previous versions of Polar SWIFT or the motivation for the current work.*

We rephrased the abstract to make our motivation more clear (see comment lines 4–5). A benefit of our approach includes the easy and self-contained coupling to a GCM. Another advantage can be that a transport parameterization based on reanalysis data and measurements can avoid deficiencies in the representation of transport in the GCMs.

There is no previous version of Polar SWIFT without the transport parameterization, see above in "general comments". The chemistry part of Polar SWIFT is only described in a few sentences, since a detailed description can be found in Wohltmann et al. (2017) and this paper focuses on the transport. We hope that the description is sufficient to put things into context.

- *Lines 65–76: It took me a couple of readings to understand what is going on – just too many variations of 'SWIFT's. Maybe an introductory sentence around Line 67 would help, stating that the transport parameterization is derived from an analysis of the total and chemical tendencies of ozone from a simulation of the ATLAS-SWIFT model?*

The complete section 3 was rewritten to consider a comment of the editor (see above). Please check if it reads better now and is more clear to the reader.

- *Lines 132–133: "For the temperature variable $\Delta T_{\text{fit}}$, we use the vortex-averaged temperature difference in a layer at a given date compared to the start date (vortex formation date, see Table 1)". Since the $\Delta T_{\text{fit}}$ term also involves subtraction of the estimate of the time-evolving radiative equilibrium temperature, as discussed a bit later, what is the reason for including the (somewhat arbitrary) temperature at the vortex formation date? Given the radiative relaxation timescale, the temperature at the vortex formation date I think would become irrelevant as an estimate of dynamical forcing with increasing time since the start date. And, as pointed out at lines 133–136 "Equation 2 suggests that the difference in temperature to the start date roughly corresponds to the deviation of the temperature on this day from the radiative equilibrium temperature by the effects of the Brewer-Dobson circulation. According to Equation 2, this would be exactly true when the temperature at the start date would be the radiative equilibrium temperature." And, of course, there is no guarantee that the temperature at the start date will be the radiative equilibrium temperature.*

  This has been largely resolved by the new method for obtaining the fit, which does not need the temperature difference to the vortex formation date anymore. Now, only the differences from day to day are used.

  We also have added the following sentence to the description of Equation 4 (previously Equation 2 in the original manuscript): "On short time scales, the change in temperature is directly correlated to the corresponding downwelling in that time period ($\overline{T}_R$ terms cancel out)."

  We agree that temperature will lose its memory to transport effects on time scales longer than the radiative relaxation time scale and that this could affect the quality of the results of the temperature-dependent term for long-term changes.

  We added the following discussion to the end of 3.3: "As a note of caution we have to stress here that while this method will work well for short-term changes in temperature and ozone, it might not work well for changes on a longer time scale. On longer time scales, temperature will start to lose memory of the transport in the past due to the radiative relaxation time scale of about 1 month." We have currently found no way to implement a method that will also correctly model the interannual variation of the very long-term changes in ozone.

- *Lines 221–223: "The differences between Polar SWIFT and MLS cannot be explained by the transport parameterization, since they are much larger than the differences of about 0.2 ppm between the transport parameterization and the transport term of ATLAS discussed in the last section 4.1." But in Figure 6, particularly for the Northern hemisphere the Polar SWIFT model without the transport parameterization (the blue line of*

*Figure 6) does a good job of estimating ozone for the two cold years. Is the argument that there should always be some positive contribution from transport so that the chemistry-only simulation should be even lower than it is, particularly for cold years? Do the authors have any reason to believe the chemistry parameterization underestimates the amount of ozone chemical destruction?*

This is a good point. This probably should be phrased more carefully. This statement implicitly relied on the assumption that the transport is represented well in the ECMWF ERA5 data and in the transport scheme of ATLAS, which, however, can't be guaranteed.

And actually, you are right that there is not really reason to believe from Wohltmann et al. (2017) that the chemistry parameterization underestimates the amount of chemical ozone destruction in cold winters (more than in warmer winters). There is just no clear indication from the results in this paper (compare Figure 15 from Wohltmann et al. (2017) to this study). The problem is also much less pronounced at other levels.

Changed the text in the manuscript to "The model overestimates ozone at 54 hPa by about 0.7–0.8 ppm compared to the MLS measurements in two winters with low ozone values (2010/2011 and 2019/2020). That is, in cold winters with large ozone depletion and a weak Brewer-Dobson circulation, while warmer years are simulated relatively well (however, the overestimation in cold winters is much less pronounced at the levels 3–5, see Figure S24 in the supplement). The differences in cold winters between Polar SWIFT and MLS at 54 hPa might not be explained by the transport parameterization alone, since they are much larger than the differences of about 0.2 ppm between the transport parameterization and the transport term of ATLAS discussed in Section 4.1 (however, this relies on the assumption that the transport is represented well in the ECMWF ERA5 data and in the transport scheme of ATLAS). Hence, the differences between Polar SWIFT and MLS in cold winters could also be a deficiency of the chemistry model of Polar SWIFT. However, there is no clear indication from Wohltmann et al. (2017) that this could be the case. A detailed discussion of the chemical model of Polar SWIFT is outside the scope of this paper."

We have to admit that it is tempting to just assume that there was almost no transport in cold years, which could explain the good agreement of Polar SWIFT without transport and MLS in these years. However, there are too many unknowns here to constrain things enough to come to a certain conclusion here.

A minor source of uncertainty will also be uncertainties in the MLS measurements, which we did not mention because they are probably small compared to other uncertainties.

---

## Author Comment (AC3)

Dear reviewer,
many thanks for your helpful comments!

**Important note**

There is a major change unrelated to the reviewer comments: To be able to resolve a comment of the editor, we had to change the method of obtaining the fit coefficients. The editor asked for error bars in Table 2 (Table 1 in the revised manuscript). Unfortunately, it was mathematically not obvious how to obtain these values with the method used in the original manuscript. For this reason, we introduced 2 major changes to the method:

1. The fit is now based on the non-accumulated ozone time series (i.e., changes per day) and not on the accumulated time series.

2. Instead of fitting the parameters for each individual year and then averaging the fit parameters over the years, we now concatenate the time series of all years before the fit to obtain a single fit parameter.

These changes prompted the following changes to the manuscript:

1. A complete rewrite of section 3 to reflect the changes in the method.

2. Addition of new Figures 2 and 4 (figure numbers from new manuscript) and deletion of old Figures 2 and 3 (figure numbers from original manuscript).

3. The contents of all figures and the numbers given in the text (fit parameters etc.) have changed sligthly throughout the paper.

The fit parameters obtained by the new method are very similar to the fit parameters of the old method. That means that the results do not change qualitatively and that the conclusions remain the same.

The new method is more elegant and gets rid of some (partly arbitrary) assumptions of the old method. However, a disadvantage is that the actual fit does not "look" as clear and intuitive in new Figure 2 as this was the case with old Figures 2 and 3.

**Note on dates for vortex formation and breakup**

During the preparation of the revised version, we noticed some slight inconsistencies in the definition of the vortex formation and breakup dates. A few dates were not consistent with the 15 million $km^2$ criterion (see "tracked changes" version of Table 3), and the validation in Figure 6 and 7 of the revised manuscript did not use exactly the same dates as the fit. This has been corrected. In addition, we changed the vortex formation date for the southern hemisphere from 1 May to 15 May to exclude some time periods with a weak vortex.

**Note on plots**

For technical reasons, we had to change the software used to create the plots. That means that colors, font sizes, axis tick marks etc. may have changed.

**General comments**

- *I am a bit surprised that the models implemented Polar SWIFT (e.g., ECHAM6, AFES4.1, ICON-NWP, ATLAS) do not have a usable scheme for tracer transport and mixing in the stratosphere. Is it because the model top is too low to resolve the stratosphere, or their scheme is so inefficient that adding four tracers will make the model unusable? I suggest the authors provide enough information about this, for example the cost of carrying four additional tracers. Otherwise, their motivations of this work are not clear to me.*

  We have changed the statements in the abstract to reflect our motivation in a better way. The original formulation may have been somewhat misleading. All of the models mentioned in the introduction have a tracer transport scheme, and we did not want to imply that these schemes can't be used. It is perhaps better to say that our scheme can be used as an alternative to the schemes for tracer transport and mixing that usually exist in GCMs, and that our scheme may have benefits.

  The initial idea of Polar SWIFT was to develop a fast and self-contained module to determine polar ozone depletion, with the aim of an easy and straightforward coupling of this module to a GCM. The concept of parameterizing the transport was our first approach, because it kept the technical interface between SWIFT and the GCM very simple. We have added this as an additional motivation to the abstract.

  There may be better methods of simulating the transport of stratospheric ozone than our parameterization and these methods are successfully used in existing models (see e.g. the models and validation in Dietmüller et al, doi:10.5194/acp-18-6699-2018). However, since our transport parameterization is fitted to reanalysis data based on measurements, it may actually perform better than the transport scheme in an existing GCM, which may e.g. suffer from deficiencies in the gravity wave parameterization that influence the Brewer-Dobson circulation in the model. This was one motivation for our parameterization that we state now more clearly in the abstract.

  For instance, we implemented tracer transport for SWIFT in ECHAM6. ECHAM6 (and also the AFES GCM) is a hydrostatic model and the tracer transport is based on a Lin-Rood scheme (Giorgetta et al., 2014, https://mpimet.mpg.de /fileadmin /publikationen /Reports /WEB_BzE_135.pdf). We tested Polar SWIFT in ECHAM6 with tracer transport and found that the tracer transport of ECHAM6 overestimated the ozone concentrations inside the vortex, especially in the southern polar vortex. The

results obtained by the transport parameterization actually were an improvement over the version with tracer transport. A reason for the bad performance of the tracer transport may be the overestimation of horizontal transport, which is a known issue in ECHAM6 (Stevens et al. 2013, doi:10.1002/jame.20015). A GCM with a more advanced tracer transport scheme (e.g. ICON) and improved vertical wave propagation will certainly compensate for some of these deficiencies.

The computational cost of adding more tracers to the GCM was not a serious issue. While the running time increased somewhat, this was not the main bottleneck in the computation.

Temperature biases in the GCM might influence the transport parameterization via the temperature dependent term. This issue and its solution are addressed in the SWIFT coupling paper, which is currently in preparation.

We do not expect that the model lid poses an issue. The GCMs that Polar SWIFT was coupled to have a model top at 1 Pa, which covers the domain of the Brewer-Dobson circulation.

Note that ATLAS is not a GCM and therefore was not included in our statement that there is no usable tracer scheme in some GCMs. ATLAS actually has a dedicated scheme for tracer transport in the stratosphere, and this scheme has been used in Wohltmann et al. (2017) for the validation runs of the chemistry part of the SWIFT model.

- *How are the tracers handled outside the polar vortex in these models? Are they advected?*

  The ozone values outside the polar vortex are taken from the internal ozone climatology of the GCM, which varies with season. Tracers are not advected outside the polar vortex. There is no interpolation applied between the two domains, since the edge of the polar vortex often forms a strong barrier between air masses and strong gradients in species concentrations are common.

  We have added discussion along these lines in an additional short section on the GCM implementation in the introduction.

- *When more efficient tracer advection scheme is available, GCMs do not need the transport parameterization. I am not familiar with the models mentioned in this manuscript, but the GCMs I have experience with all have tracers (trace gases, aerosols, and artificial tracers) transported and mixed in the stratosphere. A couple of years ago, one additional tracer roughly adds 1% overhead computational cost to the atmospheric model I used. It was a little slow, but still usable if one can keep the tracer number relatively small. Recently with the latest breakthrough of more efficient tracer advection scheme (e.g., Bradley et al., 2019; 2021), the MPI communications can be reduced by 9x in the model I use. Adding ≈30 tracers only costs ≈15% overhead. So, the tracer advection is not the primary bottleneck anymore (at least for some GCMs). If possible, I*

*would suggest the models adopt more efficient advection schemes, which provides a better solution in the long term. The method here may be used as a temporary fix when needed.*

We agree that tracer transport is not necessarily a computational bottleneck for GCM simulations. The computational cost of tracer transport was not our main motivation to develop the transport parameterization (see above). A main motivation for the transport parameterization was an easy-to-use and self-contained model. Another important motivation was to improve on the quality of the tracer transport in GCMs, which sometimes has deficiencies as described for ECHAM6 above, related e.g. to problems with the gravity wave parameterization that affect the Brewer-Dobson circulation. The fact that our model is based on a fit to reanalysis data might improve on this situation. We are also aiming to use tracer transport schemes in the future, but more validation work needs to be done.

**Minor comments**

- *L1, what does SWIFT stand for?*

  In the original manuscript (Rex et al., 2014, doi:10.5194/acp-14-6545-2014), which introduced SWIFT as a "proof-of-concept" model, SWIFT stood for "Semi-empirical Weighted Iterative Fit Technique". However, the method used for the fit in the proof-of-concept version was already replaced by a different method in the first operational SWIFT version (Wohltmann et al., 2017). Therefore, we would suggest not to consider SWIFT as an acronym, since we feel spelling out the original acronym would only cause confusion.

- *L37-38, why these 5 levels? Is that where the polar vortex is simulated? Some clarifications are helpful for the readers to understand the choice.*

  This has historical reasons. The levels roughly encompass the vertical range in which ozone depletion is observed (see Wohltmann et al., 2017). The choice of the pressure levels was guided by the pressure levels of the ECHAM (EMAC) model in this altitude range, which was the first model in which Polar SWIFT was implemented (see also Wohltmann et al., 2017). We have added some short discussion and Wohltmann et al. (2017) as a reference to section 2.1.

- *L49, do you mean a single number of ozone change rate from Polar SWIFT is used for a given layer? It is not clear if "the rate of change of ozone calculated by Polar SWIFT" is a single number or different for the grid boxes in the same layer. I feel it's a single number as SWIFT calculates the vortex-averaged value, but the sentence is a bit unclear. Please clarify.*

  The rate of change is a single number. We have added "vortex-averaged" to the sentence to make this more clear.

The sentence continues to read: ". . . for a given layer to the ozone value of every air parcel inside the vortex and inside this layer. Note that this means that the ozone field does still vary across the vortex.". We hope this clarifies that the ozone field itself (compared to the rate of change) is not a single number, but different for every air parcel. Note that ATLAS is a Langrangian model and not a Eulerian model. There are no grid boxes, but individual air parcels (locations of trajectories).

We have also added some explanation that the initialization and the transport can lead to variability in the ozone field itself.

- *Figure 6b: years 2010-2012 show a large increase in ozone observation but totally missed by the models. Do you know why?*

This is a good and interesting question. Looking at the time series of the ozone holes in the past, these winters do not seem to stand out (e.g. Bodeker and Kremser, doi:10.5194/acp-21-5289-2021). While this is interesting, we think a thorough investigation of the interannual variation of ozone in the southern hemispheric vortex and why this is not reflected in our model is outside the scope of this technical paper. May be part of the answer is that we are looking at values only from a particular day and a particular level, which may increase the variability of the measurements (and add some uncertainty to the comparison).

- *L227-230, the authors listed almost all the possible causes of why the model failed to capture the mean ozone at southern hemisphere as observed. This is not very helpful. Can the contributions from these potential factors be somewhat quantified?*

It is very hard to disentangle these effects, because they add up in the end results, without the possibility of a clear attribution. E.g., a major problem here is that it is not clear how well the transport in ECMWF compares to reality. There are ways to investigate this further (comparing to conserved tracers like N2O in runs of the full ATLAS model, for instance, or comparing the results of the chemistry of Polar SWIFT to the full chemistry model), but we think a thorough investigation of this is a lot of work and is probably outside the scope of this technical paper. We know from ATLAS run that in some circumstances, tracers like N2O or CH4 cannot be reproduced as nicely as we would like it.

Figure 16 of Wohltmann et al. (2017) shows an ozone bias at 46 hPa for the mean ozone values for a Polar SWIFT run with the full transport scheme of ATLAS that is similar to the bias observed at 41.6 hPa for run with the transport parameterization (Figure S49 of the supplement). This points into the direction that the transport parameterization is not a likely cause for the differences. We have added discussion along these lines to the manuscript.

---

## Author Comment (AC4)

Dear reviewer,
many thanks for your helpful comments!

**Important note**

There is a major change unrelated to the reviewer comments: To be able to resolve a comment of the editor, we had to change the method of obtaining the fit coefficients. The editor asked for error bars in Table 2 (Table 1 in the revised manuscript). Unfortunately, it was mathematically not obvious how to obtain these values with the method used in the original manuscript. For this reason, we introduced 2 major changes to the method:

1. The fit is now based on the non-accumulated ozone time series (i.e., changes per day) and not on the accumulated time series.

2. Instead of fitting the parameters for each individual year and then averaging the fit parameters over the years, we now concatenate the time series of all years before the fit to obtain a single fit parameter.

These changes prompted the following changes to the manuscript:

1. A complete rewrite of section 3 to reflect the changes in the method.

2. Addition of new Figures 2 and 4 (figure numbers from new manuscript) and deletion of old Figures 2 and 3 (figure numbers from original manuscript).

3. The contents of all figures and the numbers given in the text (fit parameters etc.) have changed sligthly throughout the paper.

The fit parameters obtained by the new method are very similar to the fit parameters of the old method. That means that the results do not change qualitatively and that the conclusions remain the same.

The new method is more elegant and gets rid of some (partly arbitrary) assumptions of the old method. However, a disadvantage is that the actual fit does not "look" as clear and intuitive in new Figure 2 as this was the case with old Figures 2 and 3.

**Note on dates for vortex formation and breakup**

During the preparation of the revised version, we noticed some slight inconsistencies in the definition of the vortex formation and breakup dates. A few dates were not consistent with the 15 million $km^2$ criterion (see "tracked changes" version of Table 3), and the validation in Figure 6 and 7 of the revised manuscript did not use exactly the same dates as the fit. This has been corrected. In addition, we changed the vortex formation date for the southern hemisphere from 1 May to 15 May to exclude some time periods with a weak vortex.

**Note on plots**

For technical reasons, we had to change the software used to create the plots. That means that colors, font sizes, axis tick marks etc. may have changed.

**Main comments**

Application and implementation of the method:

- *What type of GCM will benefit from this parametrization?*

  There are two "types" of GCMs that will benefit from the parameterization: a) GCMs that have no explicit tracer transport scheme (i.e., there must of course be a scheme in the dynamical core to transport water vapor etc., but there may be GCMs that do not have an easily accessible public interface for the transport of chemical or conserved tracers). b) GCMs that do have a tracer transport scheme, but have deficiencies in their representation of transport, like deficits in the representation of the Brewer-Dobson circulation caused by the gravity wave parameterization or excessive mixing. A transport parameterization based on reanalysis data and measurements like our scheme may actually perform more realistically and lead to better results than the transport of the GCM in these cases.

  Another benefit of this approach includes the easy and self-contained coupling to a GCM.

  Actually, these important motivations were missing from the manuscript, and we now state that prominently in the abstract.

  *Why do the GCMs mentioned in the paper require a transport parametrization to be able to use the Polar SWIFT ozone?*

  We have changed the statements in the abstract to reflect our motivation in a better way. The original formulations may have been somewhat misleading.

  "Require" is perhaps not the right word, since the GCMs mentioned in the manuscript have interfaces to schemes for tracer transport. It would be better to say that that our scheme can be used as an alternative to the schemes for tracer transport and mixing that usually exist in GCMs and that GCMs can benefit from the transport parameterization, or simply that they use the parameterization as a default. We may have triggered this comment by some unfortunate wording in the abstract and introduction, and have now rephrased parts of the paper (see also comment above).

  To give an example, we implemented tracer transport for SWIFT in ECHAM6. ECHAM6 (and also the AFES GCM) is a hydrostatic model and the tracer transport is based on a Lin-Rood scheme (Giorgetta et al., 2014, https://mpimet.mpg.de /fileadmin /publikationen /Reports /WEB_BzE_135.pdf).

We tested Polar SWIFT in ECHAM6 with tracer transport and found that the tracer transport of ECHAM6 overestimated the ozone concentrations inside the vortex, especially in the southern polar vortex. Actually, the results obtained by the transport parameterization were an improvement over the version with tracer transport. A reason for the bad performance of the tracer transport may be the overestimation of horizontal transport, which is a known issue in ECHAM6 (Stevens et al. 2013, doi:10.1002/jame.20015).

- *How do GCMs using this method consider ozone outside the polar vortex?*

  The ozone values outside the polar vortex are taken from the internal ozone climatology of the GCM, which can vary with season. Tracers are not advected outside the polar vortex. There is no interpolation applied between the two domains, since the edge of the polar vortex often forms a strong barrier between air masses and strong gradients in species concentrations are common.

  We have added discussion along these lines in an additional short section on the GCM implementation in the introduction.

- *How can be the limited number of vertical levels considered by this parametrization be enough for modern GCMs to simulate realistic stratospheric ozone links with meteorological variables? This is particularly concerning in the case of ICON-NWP which is not a climate model but an NWP model.*

  The main question here is not what the vertical resolution of the GCM is, but what are the typical vertical scales on which the vortex-averaged temperature profile, downwelling profile, ozone profile and ozone depletion profile vary. The vertical difference between the SWIFT pressure levels is about 2 km (expressed in altitude differences assuming typical stratospheric temperatures). Both vortex-averaged temperature and the vortex-averaged ozone profile and ozone loss profile typically only vary on a scale of several km (see e.g., Wohltmann et al., 2020, doi: 10.1029/2020GL089547, Fig. 4, Manney et al., 2011, doi:10.1038/nature10556, Fig. 3 and 4). We hope it is clear from these figures that applying SWIFT will lead to reasonable results independent from the vertical resolution of the GCM, even though there is only a limited number of SWIFT levels.

  It is an easy task to linearly interpolate the fit parameters from the two enclosing levels of SWIFT to a given level of the GCM if needed, and to apply the equations of SWIFT directly at this intermediate level. Since SWIFT is sufficiently fast, there is no computational bottleneck which prevents to use Polar SWIFT in models with a high vertical resolution.

  Considering ICON-NWP, we think the officially announced strategy of the ICON developers is to develop a "seamless" model that can also be used as a climate model. There is a "climate version" of ICON, but to our knowledge it is planned to discontinue the development of this version.

Considering these facts, it is debatable that ICON-NWP is not a climate model (despite the name).

- *More details about the implementation of the parametrization in GCMs should be added to Section 3.3*

  We have added additional discussion to the introduction to give a general idea about the parts of the implementation into a GCM that are common to all models (we thought that would be a more appropriate place than after section 3.3).

  However, a discussion of the implementation into the GCMs was actually not our main aim, and is largely outside the scope of this technical paper. Implementation will be different for different GCMs, and will also involve the chemistry part of Polar SWIFT, which is why we think a more thorough discussion leads too far away from the focus of this paper. We would like to keep the discussion short. More details for the individual models will be given in the papers in preparation mentioned in the introduction.

Validation and Comparison:

- *Comparison of the parametrization results against observations is limited to one figure in the manuscript, and the comparison against ATLAS full-chemistry or ATLAS-SWIFT is also limited. This type of comparisons should be included across figures for a clearer quantitative assessment of the parametrization performance.*

  It was a deliberate decision to limit the comparison against observations to one figure and a relatively short discussion. A comparison to observations is in large parts a validation of the chemistry part of Polar SWIFT. This manuscript is about the parameterization of transport in SWIFT, and it would be rather odd to spend a large part of this manuscript on the validation of the chemistry scheme, which is not even described in this manuscript. The same reasoning is even more true for comparisons between the full chemistry scheme of ATLAS against the simplified chemistry scheme of SWIFT.

  A validation of the chemistry scheme can be found in Wohltmann et al. (2017). We acknowledge the potential criticism that the validation is also rather limited in Wohltmann et al. (2017). But to remedy this shortcoming, it would be better to have a separate manuscript that focuses on the validation of the complete SWIFT model (or a comparison of fast ozone schemes for GCMs, maybe).

- *A more quantitative discussion of comparison results should be included in the discussions, at several points the manuscript does this only in a vague qualitative way (see specific comments below).*

  We have significantly expanded the discussion in several locations, see specific comments.

**Specific comments**

- *Abstract: It needs rewriting to summarize more clearly the work discussed here. In its current version it reads more about Polar SWIFT itself than about the transport parametrization described in this paper.*

  We have rephrased the abstract to put more emphasis on the transport parameterization and to better illustrate the motivation for this study. However, it makes sense to give a very short introduction into the complete Polar SWIFT model to set things into context and for the reader unfamiliar to the topic.

- *Lines 12–14: These studies on ozone linear schemes that should also be included here: McCormack et al (2006), Monge-Sanz et al (2011; 2022). The latter showing results from implementation in ECMWF runs and improvement over the Cariolle's scheme performance.*

  Added McCormack et al. and the Monge-Sanz papers. This wasn't meant to be an exhaustive list, given the fact that this is only a paper on a limited aspect of the SWIFT model and that a more comprehensive review of fast ozone schemes would be better suited to a study of the complete SWIFT model (including chemistry).

- *Lines 14–15: This manuscript does not compare the performance of Polar SWIFT to the mentioned schemes, and neither did Wohltmann et al. (2017). Therefore, the sentence needs to be rewritten for instance as "Polar SWIFT was developed as an alternative to these schemes...", otherwise a comparison to those schemes' performance would need to be provided.*

  Many of these schemes were originally designed only for extrapolar ozone (e.g., Cariolle and Deque, 1986; McCormack et al., 2006) or only include a simplified approach to model polar ozone depletion (e.g., Cariolle and Teyssedre, 2007; Hsu and Prather, 2009). We have added discussion along these lines to the manuscript.

  For the reasons stated above, we would like to keep the wording "improve on" in the sentence in question. Polar SWIFT is not an alternative to these schemes, but an extension that can be used in addition to an extrapolar scheme to add polar ozone chemistry to GCMs. This is hopefully made clear by the wording "improve [...] for polar ozone chemistry".

  A comparison of the chemistry in these schemes is certainly desirable and interesting (as far as they even contain a treatment of polar chemistry, that is), but is out of the scope of this paper. This technical manuscript deals with a limited aspect of the complete SWIFT model and does not deal with the chemistry of ozone. Such a comparison would be better suited to a paper dealing with the complete SWIFT model or a general validation or review paper.

- *Line 18: does the ICON-NWP model require a transport parametrization in the stratosphere to account for the evolution of the polar vortex?*

*This sentence makes the reader assume it does, better rephrase and clarify please.*

The sentence reads that "Polar SWIFT and the transport parameterization have been implemented into [. . . ] ICON". It is not implied in this sentence that the transport parameterization is required, only that there is a version were the parameterization actually is implemented. Of course, as an alternative, the tracer transport scheme of ICON could have been used.

The major changes in the abstract (e.g., ". . . can be used. . . ") hopefully make our motivation more clear and that we don't consider the transport scheme as "mandatory" in these models.

- *Lines 26–27: Changing preposition "by" to "from" would make the sentence clearer, same at the end of the sentence for "from Polar SWIFT".*

  Changed.

- *Lines 37–38: Authors need to further justify how these 5 levels can provide enough information on polar ozone for global GCM runs. Or explain what limitations there will be for GCM runs using this scheme.*

  See reply to general comment above.

- *Line 51: Rewriting as "The vortex-averaged concentrations for these species..." would be clearer.*

  Changed.

- General comment on next three comments (in addition to the specific replies): Line 53, lines 54–56 and lines 61–64.

  The chemical initialization is not discussed in detail in the manuscript, since the paper focuses on the transport and the chemical initialization (except for ozone) has only a second-order effect on the results. In particular, that means that we think that a detailed discussion of the reasoning behind the initialization leads too far away from the focus of the paper. Therefore, we would suggest not to discuss the initialization in detail in this manuscript. This was also not discussed in more detail in the paper dealing with the chemistry part of SWIFT (Wohltmann et al., 2017), and it would look a little bit odd to discuss this in more detail in the paper on the SWIFT transport than in the chemistry paper. We would agree that this could very well be seen as a deficit of the Wohltmann et al. (2017) manuscript, but it seems that this slipped through our attention and the attention of the reviewers at that time. Maybe a future updated paper on the complete SWIFT model would be more appropriate for a more detailed discussion of this.

- *Line 53: Do you use daily climatologies or seasonal climatologies? Are the daily initializations done with the corresponding seasonal climatology, am I understanding correctly? Please clarify and rewrite.*

It was stated in lines 53, 54 and 56 of the original manuscript that the climatologies are seasonal. As stated in line 55 of the original manuscript, the climatologies are a function of month. We added to the text that the climatologies are interpolated in time to the date in question.

We have added the years that the MLS and ATLAS climatologies are based on (2005–2011 and 2005–2006). Maybe this also helps to avoid confusion.

- *Lines 54–56: How is mixing climatologies from observations and model runs affecting self-consistency and how do you deal with this?*

In an ideal world, it would of course be best to take all starting values either only from observations or from the same model runs, with a preference on observations. However, this is not always possible. In this sense, the question how to deal with this may lead into the wrong direction, since the options of dealing with this are usually limited, and there is also limited knowledge with respect to the measurements of many species. However, it was exactly the aim of our approach to be as self-consistent as possible in the initialization.

We will now give a detailed account of the reasoning behind the initialization: We decided that it would be best to initialize as many species as possible from measurements. However, in our case, the sparsity of $ClONO_2$ observations prevents an initialization of this species from measurements in a larger domain. But usually, $Cl_y$ is known relatively well from measurements, since it can be deduced from tracer-tracer relationships with well-known species like $CH_4$ or $N_2O$. That also means that $Cl_y$ in the ATLAS model usually compares relatively well with measurements. For this reason, we could use the difference of $Cl_y$ and MLS HCl to obtain $ClONO_2$. It is usually a good assumption in the considered time periods and altitude range that $Cl_y$ is mainly composed of HCl and $ClONO_2$. There are now two options: Either we take MLS HCl measurements combined with ATLAS $Cl_y$ to obtain $ClONO_2$ or we take all species from the ATLAS model results. In our case, we decided it would be more consistent to use only the ATLAS values for the chlorine species, since that guarantees that HCl and $ClONO_2$ are consistent with $Cl_y$. For instance, when using HCl from MLS, it could happen that there is more HCl than $Cl_y$, which would imply negative $ClONO_2$ mixing ratios. Of course, this comes with a price: One could argue that it is more important to obtain HCl from measurements (which will differ somewhat from ATLAS HCl) than to get $Cl_y$ and chlorine species consistent. But there has to be a decision made here, which is arbitrary up to a point, since the knowledge from measurements is limited.

On a final note, it should be mentioned that ATLAS results in early winter generally compare quite well to observations (see earlier ATLAS papers as Wohltmann et al., 2013, doi:10.5194/acp-13-3909-2013 or Wohltmann et al., 2017, doi:10.5194/acp-17-10535-2017 and in particular their supplements). Of course, this is a prerequisite for an initialization with ATLAS

data.

- *Lines 61–64: if the parametrization is derived from full chemistry ATLAS and MLS observations, it is not fully clear to me why additional consideration of long-term change in chlorine content needs to be included. Please include further explanation.*

Since the climatologies are long-term averages and are not measurements for the day and year of initialization, they can only contain the average $Cl_y$ of the years that are in the average. We hope that was clear from the text in the original manuscript. We have now also added information on the years that the climatologies are based on, which may help to clarify this. To be able to use SWIFT for a particular year, and particularly for years that are not in the climatology, the chlorine content has to be scaled to match the chlorine content of that year.

- *Line 66: As earlier in the paper, this limited number of vertical levels needs further explanation and how this can affect the parametrization performance in a global GCM needs to be discussed.*

See reply to general comment above.

- *Line 79: Please quantify this statement. How small is the sensitivity to the choice of vortex dates?*

This statement was based on several runs of the fit procedure where we experimented with the vortex formation and breakup dates (typically changing them by 10 days or less) before we settled with the dates given in the table. The statement was based on the observation that usually, the fit parameters would only change by a few percent and that the plots for the fits would look virtually identical. However, we did not keep the old results.

It is also suggested by visual inspection of Figures 2 and 3 from the original manuscript and the corresponding figures in the supplement that shortening the time period of the fit in a given year will lead to similar results for the fit parameters. Of course, the same is not necessarily true when extending the time period. It has to be avoided to include time periods where the vortex clearly was not in existence. For this reason, we have chosen a rather conservative criterion here (15 million $km^2$ vortex area) to make sure that the vortex clearly existed at the dates used for the fit.

These arguments are either qualitative or our old quantitative results do not exist anymore. For this reason, we did a sensitivity run where we deliberately changed the vortex formation and vortex breakup dates. There is an almost infinite number of ways to do this, and we can only give an example here. For the example, we increased the vortex formation date by 10 days and decreased the vortex breakup date by 10 days. The following 3 tables are: 1. Table 2 from the manuscript for convenience, 2. the results for the sensitivity run, 3. percentage difference of the fit parameters

between the two cases. The last table shows that the difference is usually within a few percent. The largest difference in the table is 14.8%. We have now added to the manuscript "changing the dates within $\pm 10$ days does change the fit coefficients by less than 10% typically". We think a more thorough discussion of this in the manuscript is not needed.

| $p$ [hPa] | 69.66 | 54.04 | 41.60 | 31.77 | 24.07 | Unit |
|---|---|---|---|---|---|---|
| NH ($c_{\text{const}}$) | 0.0888 | 0.1050 | 0.1068 | 0.0969 | 0.0793 | $\cdot 10^{-7}\text{day}^{-1}$ |
| SH ($c_{\text{const}}$) | 0.1338 | 0.4850 | 0.7423 | 0.9217 | 0.9539 | $\cdot 10^{-8}\text{day}^{-1}$ |
| NH ($c_{\text{T}}$) | 0.2814 | 0.2841 | 0.2221 | 0.1489 | 0.0579 | $\cdot 10^{-7}\text{K}^{-1}$ |
| SH ($c_{\text{T}}$) | 0.2533 | 0.3097 | 0.3152 | 0.2775 | 0.1375 | $\cdot 10^{-7}\text{K}^{-1}$ |

| $p$ [hPa] | 69.66 | 54.04 | 41.60 | 31.77 | 24.07 | Unit |
|---|---|---|---|---|---|---|
| sens NH ($c_{\text{const}}$) | 0.0893 | 0.1086 | 0.1110 | 0.1015 | 0.0880 | $\cdot 10^{-7}\text{day}^{-1}$ |
| sens SH ($c_{\text{const}}$) | 0.1140 | 0.5088 | 0.7424 | 0.8794 | 0.9633 | $\cdot 10^{-8}\text{day}^{-1}$ |
| sens NH ($c_{\text{T}}$) | 0.2859 | 0.2809 | 0.2152 | 0.1493 | 0.0630 | $\cdot 10^{-7}\text{K}^{-1}$ |
| sens SH ($c_{\text{T}}$) | 0.2379 | 0.3101 | 0.3079 | 0.2656 | 0.1377 | $\cdot 10^{-7}\text{K}^{-1}$ |

| $p$ [hPa] | 69.66 | 54.04 | 41.60 | 31.77 | 24.07 |
|---|---|---|---|---|---|
| % diff NH ($c_{\text{const}}$) | 0.6139 | 3.4002 | 3.8914 | 4.7502 | 10.9429 |
| % diff SH ($c_{\text{const}}$) | 14.7707 | 4.9211 | 0.0193 | 4.5892 | 0.9925 |
| % diff NH ($c_{\text{T}}$) | 1.6027 | 1.1219 | 3.1488 | 0.2633 | 8.8086 |
| % diff SH ($c_{\text{T}}$) | 6.0820 | 0.1321 | 2.3152 | 4.2810 | 0.1140 |

- *Line 83: The date for the closure of the Antarctic ozone hole shows high interannual variability, how is this vortex breakup fixed date for all years going to affect the parameterization performance in the SH for a GCM?*

  It is important to note here that this is not the vortex breakup date used in actual model runs, but the vortex breakup date used for the fits to obtain the parameterization. As long as the actual vortex breakup date in reality in the individual years is later than 31 October, we are fine and expect no major influence on the results. In almost all cases, vortex breakup in the observations is later than end of October (e.g., Bodeker and Kremser, 10.5194/acp-21-5289-2021, Fig. 4), so that we are on the safe side.

  The only influence that the missing time periods until the actual vortex breakup in the individual years could have is that transport could behave markedly different in these time periods and that this could affect the fit parameters for the "constant change" term and temperature-dependent term. However, there is no indication from the plots and results that this could be the case. See also reply to your comment on line 79 above and the sensitivity run.

- *Figure 1: This figure should provide context for these results by including comparison at least with another validated model, for instance with ATLAS full chemistry.*

  A comparison between ATLAS-SWIFT and ATLAS with full chemistry would mainly be a comparison of the different chemistry schemes. The

transport and mixing will be exactly the same for these two model runs. We think a discussion of the differences in species mixing ratio caused by the differences in the chemistry schemes would quite clearly be out of the scope of this paper, which mainly deals with the transport of ozone.

Of course, the results for the transport term will change between the runs, since the amount of ozone that is transported will be changed by the differences in the chemistry schemes, but we are not sure if that justifies a more in-depth discussion or gives much more insight in the uncertainties of the approach.

- *Lines 92–93: From the figure, this linear approximation only holds in the long term, for shorter timescales high non-linearity can be seen, especially in the NH but not only.*

This is exactly why the temperature-dependent term is fitted in addition to the constant term. Figure 3 of the original manuscript shows that these short-term fluctuations can be explained in large parts by corresponding fluctuations in temperature (see also Figure 4 of the revised manuscript).

See also reply on your comment to lines 120–122, paragraph about radiative relaxation time scale.

*How will this affect ozone-meteo links in a GCM operating on the shorter timescales?*

The question how these short-term fluctuations in ozone induced by transport affect ozone-meteo links is a question that is extremely hard to answer. The mutual feedbacks between ozone and meteorology can be very complex. This is an area of active research and there is no simple answer to this question. We think it is outside the scope of this technical manuscript to answer this question and that this is better suited to actual scientific studies that use GCMs with interactive ozone chemistry (not necessarily our scheme).

- *Lines 120–122: Please explain and write more clearly what you mean. It reads as if "deliberately choose to simplify" could go against "find a well-working empirical relationship". What aspects are you deliberately simplifying?*

Section 3.2 of the original manuscript (now Section 3.3 in the revised version) has been rewritten in large parts following the changes triggered by a request of the editor (see "important note"). In particular, the method to obtain the fit coefficients for the temperature-dependent term is much simpler now and is directly based on the daily changes of temperature and ozone. Please have a look if things are explained in a better way now. We hope it comes out more clearly now what simplifications have been done.

There is one thing in particular that we point out more clearly now, and that is that temperature will start to lose memory of the transport in the past due to the radiative relaxation time scale of about 1 month. That

means that while our method will work well for short-term changes in temperature and ozone, it might not work well for changes on a longer time scale. We have added a paragraph on this at the end of 3.3.

We have also added a discussion of this issue to the validation in section 4.1. Figure 5 shows that the transport parameterization has sometimes difficulties to capture the magnitude of long-term changes in ozone correctly. We now give the issue of the radiative relaxation time scale as a potential reason for this.

- *Figure 3: Results shown in Figure 3 should be further discussed in the text,...*

Figure 3 has been deleted from the manuscript following the changes triggered by a request of the editor (see "important note"). In the revised manuscript, Figure 4 shows information similar to that of former Figure 3, and Figure 5 shows corresponding validation results.

As mentioned above, Section 3.2 of the original manuscript (now Section 3.3 in the revised version) has been rewritten in large parts. While some parts of the fitting procedure (and the corresponding description) have been simplified, we have also added some additional discussion in Section 3.3. E.g., there is now some discussion on the simplifications that apply to Equation 4 (new manuscript) when $t$ is small. See again also reply on your comment to lines 120–122, paragraph about radiative relaxation time scale. Please have a look if you are more happy with the discussion now.

*...and this figure should also show a comparison with data other than the parametrization.*

It is not clear to us what you are referring to. The blue line in original Figure 3 shows the change by transport from ATLAS-SWIFT. We see no obvious comparison with other data here. The red and black line are basically just a fancy way to show vortex-averaged temperature changes in the reanalysis data (involving some results of the fit). Except for showing a different reanalysis than ERA5, there is no obvious other dataset to compare with.

The new Figure 4 does simplify what is shown, and may help with this comment. Now, the daily temperature changes from ERA5 are shown directly (no results of the fit are involved in new Figure 4).

- *Line 154: Please quantify "reasonable agreement".*

This again has been affected by the changes triggered by the request of the editor (see "important note"). This sentence now refers to the new Figure 4 and we had to change the sentence to: "The good correlation between the ozone change and the temperature change...". The correlation coefficient at 54 hPa is 0.84 for the northern hemisphere and 0.75 for the southern hemisphere. This is now stated in the text.

- *Line 159–160: If this was the reason, shouldn't it be easier for the parametrization to simulate results in the SH than in the NH? What about the non-linearity between ozone and temperature in the SH vortex?*

  Following the changes triggered by the request of the editor (see "important note"), this comment has been removed. Actually, Figure 4 of the revised manuscript shows that the correlation between the daily temperature changes and the daily ozone changes over all years is quite similar in the northern and southern hemisphere at 54 hPa. This can also be seen by the similar values of the fit parameters for the temperature-dependent term in the southern and northern hemisphere in Table 1. We have added discussion on this to the manuscript.

  The bad agreement of the temperature term of the fit to the changes in ozone that can be seen in Figure 3 (original manuscript) for the southern hemisphere in 2011 seems to be caused by changes on time scales longer than just a few days that don't quite agree between ozone and the temperature term. The fit struggles to fit the longer-term changes, at the expense of the magnitude of the shorter-term changes, which it does not get quite right. For the old method, results very much depend on the individual year in the southern hemisphere (see figures corresponding to old Figure 3 in the original supplement). The new method mitigates this effect by fitting all years at the same time.

  In our opinion, it is likely that one of the reasons for the sometimes bad agreement of the long-term changes in ozone and temperature is that temperature will start to lose memory of the transport in the past due to the radiative relaxation time scale of about 1 month, as mentioned in the reply to lines 120–122. This is now discussed in Section 4.1 when looking at the long-term changes in Figure 5.

- *Section 3.3: Is temperature the only variable needed from the GCM when using this parametrization?*

  Yes, vortex-averaged temperature is the only variable needed.

  *This seems to be indicated by Eq 3 but there is no specific mention in the text. Please add a specific description in the text...*

  We have added this information to the introduction.

  *...and add more information about implementation in a GCM.*

  We have added discussion on the implementation into a GCM to the introduction.

- *Figure 5: The comparison with ATLAS-SWIFT should also be added to previous figures, in particular Figure 3, and corresponding ones in the supplement.*

  Figure 3 has been removed from the paper, following the changes requested by the editor (see "important note"). But we are not sure if we understand the comment: The ATLAS-SWIFT curve (the blue curve in Fig. 5) is

shown both in Fig. 1 e and f of the original manuscript (blue, exactly the same curve), in Fig. 2 of the original manuscript (blue, again exactly the same curve, figure has been removed in the revised manuscript) and in Fig. 3 of the original manuscript (blue, now with the linear trend subtracted, figure has been removed in revised manuscript). Are we talking about the same thing here?

- *Lines 190–192: what about the % difference against observations?*

We have added figures to the supplement that show the difference between the cumulated vortex-averaged ozone change by transport at vortex break up between the transport parameterization and ATLAS-SWIFT divided by the simulated or observed ozone at this day (S25 and S50, right columns). These figures show that the order of magnitude of the difference to observations is similar.

*And is a 10 % order of magnitude good enough for the ICON-NWP model?*

What do you mean by "good enough"? It seems the question is a little bit unclear and therefore hard to answer. Why do you only refer to the ICON-NWP model and not to the other models?

We are not convinced that this question actually leads somewhere. Models can only use parameterizations that are the current state-of-the-art. It does not help to point out that there is a 10% uncertainty in some parameterization if you can't do better anyway. In the past, it has never stopped the modelling community from doing scientific studies that some variables and processes are even represented much worse (think e.g. of the considerable temperature biases in the stratosphere. These can reach 5–10 K in the polar vortex easily in ECHAM6, for example).

- *Line 211: Delete "The" at start of sentence.*

Changed.

- *Lines 212–213: Why is the difference so small? This needs to be further discussed, isn't the purpose of the parametrization to improve this?*

This is a good point. This was not discussed in the manuscript, although it is a crucial point, and is an obvious omission. We have now added an extended discussion to the manuscript in this paragraph.

Actually, while the temperature-dependent term captures the short-term changes in ozone quite well (see Figure 5, and Figures S19–23, S44–48), it is struggling to improve the simulation of the long-term changes in ozone. To repeat this again, this may be related to the fact that the parameterization might not work well for time periods longer than the radiative relaxation time scale. In fact, the results for the long-term changes for the transport parameterization without the temperature-dependent term do not perform worse than for the parameterization with that term on average (Figures 6, 7, S24, S49). At best, the results are inconclusive. We have calculated the root mean square error (RMSE) of the difference

of the simulated ozone of the stand-alone models at the time of vortex breakup to the MLS measurements and the correlation coefficient of the same quantities to give a more quantitative account of this. E.g., the RMSE of the parameterization with the temperature-dependent term at 54 hPa in the northern hemisphere is 0.36 ppm, while it is 0.44 ppm for the parameterization with only the "constant change" term. In contrast, the correlation coefficient is 0.88 and 0.86, respectively. Values for the RMSE and the correlation coefficients can be found in Figure 7 and in the Figures S24 and S49 in the supplement for the other levels. It is apparent that sometimes the parameterization with the temperature-dependent term performs better in terms of RMSE, and sometimes the parameterization without the term does perform better, and that the same is true for the correlation coefficient, with no clear pattern. This indicates that there are opportunities for improvement for the parameterization.

- *Figure 6: Interannual variability in the NH is far from MLS observations.*

  We have considerably expanded discussion of the interannual variability in the manuscript.

  We would disagree that interannual variability in the NH is far from MLS observations. Looking at Figure 7a (not 6a), there is actually quite a decent correlation between the MLS measurements and the results of the stand-alone model for the interannual variability at 54 hPa. The correlation coefficient at 54 hPa is 0.88 for all years and 0.82 for all years except 2010/2011 and 2019/2020. RMSE is 0.36 ppm, which is a reasonable value compared to the observed and simulated ozone values of 2–3 ppm. We state these values in the text now and have added the correlation coefficients to Figure 7 and the corresponding figures in the supplement. Your impression may have been caused by the fact that the model overestimates ozone compared to measurements in the two winters with very low ozone values (2010/2011 and 2019/2020), that is, in cold winters with large ozone depletion and a weak Brewer-Dobson circulation, while the large majority of the warmer years is simulated relatively well. This was explicitly discussed in the text in the original manuscript (lines 216–218), and the discussion has been expanded in the new manuscript.

  Note that the results depend somewhat on the SWIFT level, see figures corresponding to Figure 7 in the supplement. In particular, the overestimation of ozone in cold winters is less pronounced at other levels than 54 hPa.

  *In the SH both the interannual variability and the mean value are far from MLS observations. Please develop further the discussion on these points in the paper.*

  We would like to keep this discussion short, since it deals with the complete SWIFT model and not only with the transport parameterization. It probably leads to far away from the scope of this technical paper to add a detailed discussion here.

Concerning the mean values, it is very hard to disentangle the effects mentioned in the text, because they add up in the end results, without the possibility of a clear attribution. E.g., a major problem here is that it is not clear how well the transport in ECMWF compares to reality. There are ways to investigate this further (comparing to conserved tracers like N2O in runs of the full ATLAS model, for instance, or comparing the results of the chemistry of Polar SWIFT to the full chemistry model), but we think a thorough investigation of this is a lot of work and is outside the scope of this technical paper. We know from ATLAS run that in some circumstances, tracers like N2O or CH4 cannot be reproduced as nicely as we would like it.

Figure 16 of Wohltmann et al. (2017) shows an ozone bias at 46 hPa for the mean ozone values for a Polar SWIFT run with the full transport scheme of ATLAS that is similar to the bias observed at 41.6 hPa for run with the transport parameterization (Figure S49 of the supplement). This points into the direction that the transport parameterization is not a likely cause for the differences. We have added discussion along these lines to the manuscript.

A likely reason that is more difficult to get the variability right in the southern hemisphere is simply that the interannual variability is much lower in the southern hemisphere.

- *Lines 216–218: Please quantify these statements. By how much does the model overestimate ozone in cold winters and what does "relatively well" exactly mean for warm winters?*

We now state that the difference at 54 hPa is about 0.7–0.8 ppm and give the RMSE and correlation coefficients for the warm winters and all winters in Figure 7 and the corresponding figures in the supplement, and discuss some of these values in the text (see also comments above).

- *Lines 220–221: Then shouldn't the parameterization allow for a different vortex breakup date for warm and cold winters?*

It seems there is a misunderstanding here what was done in Figure 6 and discussed in the corresponding discussion. The parameterization does allow for different vortex breakup dates.

Actually, a more detailed description of the time periods when SWIFT is switched on and off in every winter in the model setup in the stand-alone models, in ATLAS and in GCMs is probably helpful here. For the stand-alone version in Section 4.1 and the GCMs, this description was actually missing from the manuscript, while it was given for the stand-alone model of Section 4.2 (and ATLAS-SWIFT earlier in the manuscript). A detailed discussion is also partly missing for the chemistry part in Wohltmann et al. (2017). The reason for the short discussion of this in Wohltmann et al. (2017) was that the chemistry part is not very sensitive to the exact time period. When there is no vortex or only a small vortex, there is usually

no chlorine activation and the temperatures are high, and the chemical change of ozone is small.

However, this is different for the transport parameterization. It is not desirable to add a constant amount of ozone every day when there is no vortex or only a small vortex. For this reason, we did only apply SWIFT in the stand-alone models of Section 4.1 and 4.2 in the time periods when the vortex area exceeded 15 million $km^2$ at 54 hPa. While we stated that in the manuscript for the complete stand-alone version of Section 4.2 (see lines 200–201 of the original manuscript), this information was missing for the stand-alone version of the transport parameterization in Section 4.1, and more importantly, there was also no recommendation how to implement the time periods into a GCM. We now state the time periods for the stand-alone model in Section 4.1 and have added a recommendation for the time periods in a GCM.

Figure 6 shows the results at a day shortly before vortex breakup in each individual year (see Table 3, we have added a reference to the table in the sentence). Thus, the vortex breakup date is considered in Figure 6 and the accompanying discussion. The sentence in lines 220–221 was exactly about this. To make this more clear, we now write "differences in the simulated ozone in the figure" and not only "differences in the simulated ozone" in the preceding sentence. The rationale behind showing the simulation results at the day of vortex breakup was to have a time period as long as possible in each individual winter where SWIFT is able to simulate changes in ozone, so that we get a worst-case estimate of potential systematic errors that add up during the model run. We have added discussion along these lines to the manuscript.

- *Lines 224-225: OK, but please add a reference documenting the performance of Polar SWIFT compared to observations please.*

Following a comment of a different reviewer, we have phrased this part more carefully, which may also help with your comment. It is not quite clear to us what your comment aims at. It seems that you are silently assuming that there must be a reference (apart from Wohltmann et al., 2017) that performs a detailed validation of the complete Polar SWIFT model. However, there is no such publication. Such a study would certainly be desirable, but has not been performed so far.

- *Lines 227-230: Key references should be added into this sentence documenting validity of linearity assumption, transport in ERA5 and chemistry performance of Polar SWIFT that can back up or strongly suggest these hypotheses.*

This was merely intended as a list of suggestions (with the alternative of just being silent here). There are no detailed studies on this so far. We think it is outside the scope of this paper to go into more detail here, since this mainly concerns the validation of the complete SWIFT model including chemistry, see also several of the replies above.

- *Line 234: However, these three models do have complex transport schemes. Please summarize here clearly why they need this type of transport parametrization.*

  Please also see the reply to your first and second main comments. We have now rephrased and extended the abstract to make our motivation to implement the transport parameterization into these models more clear, and repeat this shortly in the conclusions now. We do no write that the models "need" the transport parameterization, just that the transport parameterization has been implemented into these models. It would be better to say that they benefit from the transport parameterization, or simply that they use the parameterization as a default.

- *Line 238: "slightly better" needs to be quantified. The agreement with observations and ATLAS is very different in NH and SH from the results you show in the paper...*

  Changed the sentence to "Agreement of the complete SWIFT model (including chemistry) with observations is usually better in the northern hemisphere than in the southern hemisphere (see Figs. 6, 7 and supplement).", so the readers can judge by themselves by looking at the figures. We give some numbers in the conclusion now. However, we don't want to go too much into detail in the conclusions.

  *...indicating the parametrization may not be fully suitable for SH polar ozone.*

  We wouldn't phrase it this way. This is always a question of the current state-of-the-art and if one could do possibly better. We fully agree that there is room for improvement and that we would be happy if that can be improved in a new version in the future. But in cases where a full chemistry scheme is computationally too expensive, it may be a better alternative than just using a fixed ozone climatology.

- *Lines 238-240: Please add some text to summarize why it performs better in the NH than in the SH. Same for the contribution of the transport and temperature terms mentioned in the last sentence.*

  Again, we think it is outside the scope of this paper to give a detailed validation of the complete SWIFT model.

  We have no clear indication why the model performs better in the northern hemisphere. We would like to leave the text as is and only give the speculative list of reasons in lines 227–230 of the original manuscript in Section 4.2. This is certainly not easy to answer, and for the implementation into a GCM the information that it performs better in the northern hemisphere is more important than why it performs better.

  The same is true for the fact that the "constant change" term generates a larger contribution to the change of ozone than the temperature-dependent term. Part of the reason is certainly that the "constant change" represents something like the climatological mean state of the Brewer-Dobson

circulation, and then the temperature-dependent term adds some variability that is mainly based only on short-term changes. But that does not explain the exact quantitative result. For many questions like this, the answer can only be "Because it comes out like this when all the complex processes that are at work here play together", and maybe asking this does not really lead somewhere.

- *Acronyms: Most acronyms throughout the paper (abstract and main text) are not spelled out. If possible please do so on first appearance.*

  In the original manuscript (Rex et al., 2014, doi:10.5194/acp-14-6545-2014), which introduced SWIFT as a "proof-of-concept" model, SWIFT stood for "Semi-empirical Weighted Iterative Fit Technique". However, the method used for the fit in the proof-of-concept version was already replaced by a different method in the first operational SWIFT version (Wohltmann et al., 2017). Therefore, we would suggest not to consider SWIFT as an acronym, since we feel spelling out the original acronym would only cause confusion.

  For similar reasons, I have decided some years ago not to treat ATLAS as an acronym anymore, which I felt was at my discretion as the main developer.

  MLS is spelled out at first occurence now.

  In our opinion, spelling out acronyms when this does not add any additional insight only clutters the text and decreases readability. For this reason, we have refrained from spelling out most of the acronyms, in particular the acronymns of the GCMs, which are much better known under their acronym.

- *Figures S28 and S56: What percentage of the total ozone does this change represent? This information should be added to these figures.*

  We have added figures to the supplement that show the difference between the cumulated vortex-averaged ozone change by transport at vortex break up between the transport parameterization and ATLAS-SWIFT divided by the simulated or observed ozone at this day (The figure numbers have changed to S25 and S50. The plots are shown in the column on the right).

---

## Author Response (AR2)

Dear editor, dear reviewers,
here are the replies to the comments.

**Reviewer 1**

- *The revisions to Section 1, particularly over lines 34–45, make the description of what Polar SWIFT does much clearer and helps set the stage for the rest of the paper. About the only confusion I had was the introduction of Equation 3, which, given how like it is to Equation 1, made me wonder if there was a repeated section of the text. Do you really need Equation 3? Or could you just say you are using Equation 1 with the temperature change from ERA5?*

  While it may look superficially as if Equation 1 and Equation 3 are identical, they are actually not the same equation, and it would be both confusing and mathematically incorrect to refrain from including Equation 3. Equation 1 describes the application of the parameterization in the model. Equation 3 describes a linear fit model used to obtain some of the parameters in the equation. The variables have a different meaning, and they are described in the text following Equation 3. Therefore, we think the equation and the following text are necessary to be able to understand what we have done.

**Reviewer 2 (this seems to be reviewer 3 from the first round of reviews?)**

- *The information authors include in their reply about the types of GCM that benefit from this type of scheme is very clear. I do welcome the edits done to the abstract, but information on this should also be included in the main text, not only in the abstract. I suggest adding 1–2 sentences at the end of line 24 mentioning the potential benefits for these GCMs (as authors mention in their reply to my comment).*

  Added a new paragraph based on the reply to the comment.

- *Similarly, the reasons for using the transport parameterization, given by the authors as answer to the second part of my same comment, provide useful information for readers, in particular the example with ECHAM6. At least part of it would need to be included into the main text. This could be merged to the additional sentence from previous point, for instance.*

  Changed as suggested.

- *Number of vertical levels: I agree with the reasons authors give in their reply to my comment. I still think it is good to add information into the main text to justify this choice, for instance 1-2 sentences pointing to the reason and references given in the authors' reply to my related comment.*

  Changed as suggested.

- *Line 20, this sentence is still ambiguous and leads to misunderstanding. It needs to be edited to make it as clear as the reply authors provide now to my related comment. I suggest they use the same wording as they do in their reply: "Polar SWIFT is intended as an extension that can be used in addition to an extrapolar scheme to add polar ozone chemistry to GCMs."*

We changed your suggestion slightly and changed the sentence in the manuscript at line 20 to: "Polar SWIFT is intended to add a more sophisticated polar ozone chemistry scheme to GCMs." The reason for changing your suggestion is that it is not necessary to run Polar SWIFT in combination with an extrapolar scheme. It is possible as well to use a climatology outside the polar vortex. Actually, this is the way that Polar SWIFT is implemented into ECHAM and AFES in the moment.

- *Initialization, Section 2.2. As authors state in their reply, previous publications don't include information on initialization choices, it'd be good to briefly add something here to justify choices for optimizing self-consistency in the initialization. I agree a full discussion is not appropriate for this paper, but it can be done briefly mentioning the compromise and choices adopted between model data and MLS observations. My suggestion: "As many species as possible have been initialized with MLS measurements, however ClONO2 observations do not provide enough coverage. That is why we have decided to use only ATLAS values for chlorine species, since that helps guarantee consistency of HCl and ClONO2 with Cly."*

Changed as suggested.

- *Line 325: "At the moment..."*

Changed as suggested.

- *Line 325-326. I welcome the edits done to the abstract in this respect, but some clarifying words should also be added in the conclusions. I suggest briefly adding to the end of this sentence "...as an alternative to the transport options in these models." In line with the end of your reply to my specific comment for the old version.*

Changed as suggested.

---

## Author Response (AR3)

Dear editor,

- *Please ensure that the colour schemes used in your maps and charts allow readers with colour vision deficiencies to correctly interpret your findings. Please check your figure 6 and the supplement figures using the Coblis Color Blindness Simulator and revise the colour schemes accordingly.*

  Checked.

- *Further, if you applied changes to the supplement PDF, a tracked changes file is mandatory. This is why I kindly ask you to include the tracked-changes of your manuscript together with the tracked-changes of your supplement in one PDF. Considering the size of your supplement, I would suggest to only include the figures that actually have changed.*

  No changes were applied to the supplement.